# Quantum-critical properties of the one- and two-dimensional random transverse-field Ising model from large-scale quantum Monte Carlo simulations

Calvin Krämer$^\star$, Jan Alexander Koziol, Anja Langheld,
Max Hörmann and Kai Phillip Schmidt$^\dagger$

Department of Physics, Staudtstraße 7,
Friedrich-Alexander-Universität Erlangen-Nürnberg, Germany

$\star$ calvin.kraemer@fau.de , $\dagger$ kai.phillip.schmidt@fau.de

## Abstract

We study the ferromagnetic transverse-field Ising model with quenched disorder at $T = 0$ in one and two dimensions by means of stochastic series expansion quantum Monte Carlo simulations using a rigorous zero-temperature scheme. Using a sample-replication method and averaged Binder ratios, we determine the critical shift and width exponents $\nu_s$ and $\nu_w$ as well as unbiased critical points by finite-size scaling. Further, scaling of the disorder-averaged magnetisation at the critical point is used to determine the order-parameter critical exponent $\beta$ and the critical exponent $\nu_{av}$ of the average correlation length. The dynamic scaling in the Griffiths phase is investigated by measuring the local susceptibility in the disordered phase and the dynamic exponent $z'$ is extracted. By applying various finite-size scaling protocols, we provide an extensive and comprehensive comparison between the different approaches on equal footing. The emphasis on effective zero-temperature simulations resolves several inconsistencies in existing literature.

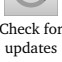

# 1   Introduction

Quantum phase transitions have been an integral subject of interest in the field of quantum many-body physics for a long time (for reviews on this topic see Refs. [1,2]). Driven solely by quantum fluctuations, they are defined as non-analytic points in the ground-state properties of quantum systems and are therefore only defined strictly at zero temperature [1,2]. Further, each quantum-critical point at zero temperature influences a finite-temperature quantum-critical region in which thermal and quantum fluctuations compete [1,2]. One of the paradigmatic models to study quantum phase transitions is the transverse-field Ising model (TFIM). The critical behaviour of the TFIM has already been intensively studied on a wide variety of geometries (for some examples see Ref. [3–7]) and various extensions like long-range interactions [8–13] or coupling to light [14,15] have been applied to this model to investigate their effect on quantum systems.

Another extension to the TFIM is quenched disorder, which refers to a random choice of couplings that are constant in time. Quenched disorder is motivated as an inevitable ingredient of any condensed matter system since variations in the couplings might occur by accident, e. g. due to imperfections in the fabrication process [16,17] or on purpose, e. g. through the replacement of atoms in a compound [18–23]. It has been found that quenched disorder has a great impact on phase transitions, in particular on quantum phase transitions [24]. In the latter, Griffiths singularities [18,19,25,26] with exponentially diverging timescales emerge more easily due to the perfectly correlated disorder in imaginary time [24]. The Harris criterion predicts whether a critical point is stable under the influence of disorder [27]. If the critical exponent $\nu$ violates the inequality $\nu > 2/d$ (with $d$ the spatial dimension of the system), the criticality is expected to change with respect to the clean system without disorder [24].

In the case of the TFIM with random transverse field and/or Ising couplings (RTFIM) on the linear chain or square lattice, the disorder is relevant and the universality class changes under the influence of disorder [28]. Moreover, due to rare strong fluctuations of the disorder, the model exhibits Griffiths singularities [25, 26] and shows activated scaling with exponentially diverging timescales [28]. From a renormalisation group (RG) point of view, this behaviour can be traced back to the fact that the critical behaviour is controlled by an infinite-disorder fixed point (IDFP) [28–30]. Beyond the one-dimensional RTFIM, no analytical results are known to study quantum criticality of the RTFIM and numerical approaches have to be employed. A method directly adjusted to the problem is the strong-disorder renormalisation group (SDRG) [28, 31–36] approach, which is used to study the RTFIM in several dimensions. This approach is well suited to study the criticality, since the SDRG scheme gets exact at the critical point if an IDFP exists [28, 30]. There are also several (quantum-) Monte-Carlo studies considering the RTFIM [37–40]. All numerical methods suffer from the fact, that averaging over a huge amount of disorder realisations is required, leading to a large computational effort. Moreover, convergence is not necessarily improved with increasing system size since the RTFIM is not self-averaging in the vicinity of the critical point [27, 41]. Additionally, finite-temperature methods, e. g. the quantum Monte Carlo studies mentioned above including this work, suffer from the fact that the typical energy scale is decreasing exponentially with the system size.

In this work, we use the stochastic series expansion (SSE) quantum Monte Carlo (QMC) method introduced by A. Sandvik for TFIMs [42–45] to investigate the RTFIM on the linear chain and the square lattice. We apply various finite-size scaling techniques in order to extract unbiased estimates for the critical point and the critical exponents. We compare these methods and work out which subtleties have to be considered for disordered systems in comparison to pure systems. Therefore, with this work we aim to provide a large scale comprehensive comparison on which finite-size scaling methods are suitable (or not) when a system is affected by disorder or more specifically when a quantum critical point is attracted to an IDFP.

In Sec. 2, we introduce the RTFIM and summarize the effect of disorder on the quantum criticality (in comparison to the pure model) by reviewing analytical results for the one-dimensional RTFIM and giving an overview of recent results on the RTFIM in two dimensions. These phenomena are understood using the theoretical concept of an IDFP. In Sec. 3, we introduce the SSE QMC method, which is used to study the RTFIM in this paper. This is followed by Sec. 4, where we present the finite-size scaling techniques that work best, which are then applied to the data in Sec. 5. Besides the primary techniques to extract the critical point and critical exponents in Sec. 5.1 and 5.2, we also investigate the temperature convergence in Sec. 5.3. Finally, we conclude the paper in Sec. 6.

## 2 Random transverse-field Ising model

The Hamiltonian of the RTFIM is given by

$$\mathcal{H} = \sum_{\langle i,j \rangle} J_{i,j} \sigma_i^z \sigma_j^z - \sum_i h_i \sigma_i^x \, , \tag{1}$$

where $\sigma_i^{z/x}$ are the Pauli matrices, describing spins 1/2 located on lattice sites $i$. The nearest-neighbour Ising coupling strengths $J_{i,j}$ and field strengths $h_i$ are sampled according to

$$J_{i,j} \in \mathcal{U}_{(-1,0)} \, , \quad h_i \in \mathcal{U}_{(x,h)} \, , \tag{2}$$

with $\mathcal{U}_{(a,b)}$ denoting a uniform probability distribution on the interval $[a, b]$, $x$ being a parameter to tune the disorder in the transverse field and the control parameter $h$ being the upper

bound to the transverse field. In this work, we consider two different types of disorder: Bond-disorder abbreviated by $h$-fix, where $x = h$, i. e. there is no disorder in the transverse field, and bond- and field-disorder abbreviated by $h$-box, where $x = 0$, i. e. $h_i \in [0, h]$ uniformly. For the ferromagnetic systems discussed in this paper ($J_{i,j} < 0 \; \forall i, j$), the RTFIM exhibits an ordered phase, when the field strengths $h_i$ are much smaller than the bond strengths $|J_{i,j}|$ and a disordered phase, when the $h_i$ are much larger than the $|J_{i,j}|$. The two phases can be distinguished by a $\mathbb{Z}_2$-symmetry. The order parameter associated with the spontaneous symmetry breaking at the phase transition is the $z$-magnetisation

$$m = \frac{1}{N} \sum_i \sigma_i^z \, . \tag{3}$$

In the thermodynamic limit, the order parameter is finite in the ordered phase and vanishes in the disordered phase. However, since we investigate the RTFIM using a finite-temperature statistical method on disordered lattices, we have to introduce several types of averaging. We denote the thermodynamic average by $\langle \ldots \rangle$ and the average over disorder configurations by $[ \ldots ]$. Furthermore in finite systems the ground state is a mixed state of positive and negative magnetisation eigenstates, which causes $\langle m \rangle$ to vanish not just in the disordered but also in the ordered phase. Therefore the order parameter we actually consider is $\langle m^2 \rangle$ for a single disorder configuration and $[ \langle m^2 \rangle ]$ for the averaged system.

Unlike the transverse-field Ising chain without quenched disorder [3, 46, 47], the RTFIM on a chain can no longer be solved analytically due to the lack of translation symmetry. Nevertheless, using a Jordan-Wigner transformation, the following condition can be derived for the bond and field strengths $J_{ij}$ and $h_i$ at the quantum phase transition [48]

$$\prod_i h_i = \prod_i \left| J_{i,i+1} \right| \, . \tag{4}$$

This relation is exact up to a neglected boundary term, that vanishes in the thermodynamic limit (see Ref. [48] or App. B). However, this equation cannot be used to make any statements about the universality class of the critical point. The Harris criterion [27] states, that the critical point of a clean system is stable against disorder, if the associated critical exponent $\nu$ is larger than $2/d$, where $d$ is the dimension of the system. If the Harris criterion is fulfilled, disorder does not affect the critical exponents. In the case of the one- and two-dimensional RTFIM, the Harris-criterion is violated which implies a change of the universality class with respect to the clean system and self-averaging is no longer provided in the vicinity of the critical point [41, 49]. The SDRG [31] technique is particularly suitable for investigating the critical behaviour of disordered systems. D. S. Fisher was able to show that the SDRG method becomes exact at the critical point if the disorder increases infinitely under renormalisation [28]. One speaks of an IDFP, where the disorder dominates statistical fluctuations coming from temperature or quantum uncertainty. Rare regions, i. e. strongly coupled spin clusters, have an $\mathcal{O}(1)$ contribution to observables even in infinite systems, so that a distinction must be made between average and typical observables [33]. Typical and average correlation lengths may also diverge differently at the critical point [29, 50], e. g. the correlation lengths can be defined as

$$\xi_{\text{av}}^{-1} = \lim_{|i-j| \to \infty} -\frac{\log \left[ \langle \sigma_i \sigma_j \rangle \right]}{|i-j|} \, , \qquad \xi_{\text{typ}}^{-1} = \lim_{|i-j| \to \infty} -\left[ \frac{\log \langle \sigma_i \sigma_j \rangle}{|i-j|} \right] \, , \tag{5}$$

where $[ \ldots ]$ denotes the average over disorder realisations. Furthermore, rare regions show singular behaviour with exponentially small energy gaps in the ordered and disordered phase also away from the critical point, leading to a Griffiths phase [25, 26]. These exponentially

small energy gaps, equivalent to exponentially long timescales, also influence the singular behaviour close to the critical point leading to an exponential dependence between the characteristic timescale $\xi_\tau$ (characteristic energy scale $\Delta$) and the correlation length $\xi$

$$\log \xi_\tau \sim \xi^\psi \quad \longrightarrow \quad \xi_\tau \sim 1/\Delta \sim \exp\left(c, \xi^\psi\right), \tag{6}$$

called activated scaling [29]. Since the dynamic critical exponent $z$ defined by the algebraic power-law $\xi_\tau \sim \xi^z$ does not comply with this definition and the exponential scaling is stronger than the common algebraic scaling, it is often referred to as $z = \infty$. For the RTFIM on a chain, analytic SDRG results state that the average correlation length diverges with an exponent $\nu_{\text{av}} = 2$, whereas the typical correlation length diverges with an exponent $\nu_{\text{typ}} = 1$ at the phase transition [28]. The order parameter critical exponent is found to be $\beta = (3 - \sqrt{5})/2$ and the critical exponent of the activated dynamic scaling is given by $\psi = 1/2$ [28]. Besides the SDRG method, there are also other works on the one-dimensional RTFIM using Monte Carlo methods [37] or free-fermion techniques [51, 52] which are in good agreement. The finite-size analysis of the distribution of pseudo-critical points $P_L(h_c)$ in Ref. [52] is discussing two more critical exponents $\nu_{\text{s/w}}$ describing the shift of the mean and the width of $P_L(h_c)$. They are determined to be $\nu_s = 1$ and $\nu_w = 2$ and identified with $\nu_{\text{typ/av}}$ [34, 52]. Furthermore, the distribution of energy gaps in the Griffiths phase can be used to define a dynamic exponent $z'$ locally, which is finite in the Griffiths phase and diverges when approaching the quantum critical point [51].

The situation is less clear for the RTFIM in two dimensions. SDRG results indicate that the two-dimensional model is also attracted to an IDFP under renormalisation [32, 34–36, 53, 54]. Note that this approach does not provide an analytical result in two dimensions (in comparison to the chain), but requires a numerical treatment [34, 35]. QMC results support the hypothesis of an IDFP by reproducing the same behaviour of the distribution of energy gaps as in the one-dimensional model [38, 39]. However, there is a recent QMC study indicating that only the RTFIM with disorder of type $h$-box is attracted to the IDFP, whereas for the model with disorder of type $h$-fix there are indications that the transition is of 2D transverse-field Ising spin glass universality [40]. The SDRG results agree well with each other and the probably most precise results for the critical exponents extracted from finite-size scaling of the distribution of critical points and the magnetic moment are $\nu_s = \nu_w = 1.24$ and $\beta = 1.22$ [35]. The energy gap compared to the correlation length is showing activated scaling with an exponent $\psi = 0.48$ [35] and the typical correlation length scales with an exponent $\nu_{\text{typ}} = 0.64$ [36]. In terms of the location of the critical point, there are different values in literature: On the one hand, SDRG predicts $h_c = 5.37$ for $h$-box-type disorder [35] (other SDRG studies are consistent with this result [36, 53, 54]). On the other hand, world-line QMC studies predict two different values $h_c = 4.2$ [38, 39] and $h_c = 7.52$ [40].

## 3 Stochastic series expansion quantum Monte Carlo

The stochastic series expansion (SSE) approach is a quantum Monte Carlo (QMC) method based on a high-temperature expansion of the partition function used to simulate a finite amount of spins at a finite temperature. The method has been pioneered by A. Sandvik [13, 42–45] and is based on the idea to lift the configuration space from solely spin states to also include so-called operator sequences. In order to set up a SSE QMC sampling, the Hamiltonian of the RTFIM has to be decomposed into a sum of operators

$$\mathcal{H} = -\sum_{i=1}^{N}\sum_{j=0}^{i} \mathcal{H}_{i,j} + c, \tag{7}$$

together with a suitable computational basis $\{|\alpha\rangle\}$, where $c$ is a constant shift that does not change the physics. The operators $\mathcal{H}_{i,j}$ must be chosen to have only non-negative entries in the matrix representation in the computational basis. In addition, the operators need to fulfil the non-branching rule, i.e. acting on a basis state does not create superpositions of basis states. A suitable way to set up a SSE QMC sampling for arbitrary TFIMs, is to choose the $z$-basis $\{|\alpha\rangle\} = \{|\sigma_1^z, ... \sigma_N^z\rangle\}$ as a computational basis and decompose the Hamiltonian into the operators

$$\mathcal{H}_{0,0} = \mathbb{1}, \qquad \mathcal{H}_{i,0} = h_i\left(\sigma_i^+ + \sigma_i^-\right), \tag{8}$$

$$\mathcal{H}_{i,i} = h_i, \qquad \mathcal{H}_{i,j} = |J_{i,j}| - J_{i,j}\,\sigma_i^z\sigma_j^z, \tag{9}$$

for $i, j \geq 1$ and $j < i$, where $\mathcal{H}_{0,0}$ is introduced for algorithmic reasons only and is not part of $\mathcal{H}$. In the SSE approach, the partition function is expanded in powers of $\beta\mathcal{H}$, the trace over the operators is executed in the chosen computational basis $\{|\alpha\rangle\}$ and the decomposed Hamiltonian is inserted into the partition function

$$Z = \mathrm{Tr}(e^{-\beta\mathcal{H}}) = \sum_{\{|\alpha\rangle\}} \sum_{n=0}^{\infty} \frac{\beta^n}{n!} \langle\alpha| \Big(\sum_{i=1}^{N}\sum_{j=0}^{i} \mathcal{H}_{i,j}\Big)^n |\alpha\rangle \tag{10}$$

$$= \sum_{\{|\alpha\rangle\}} \sum_{n=0}^{\infty} \sum_{\{S_n\}} \frac{\beta^n}{n!} \langle\alpha| \prod_{l=1}^{n} \mathcal{H}_{i(l),j(l)} |\alpha\rangle. \tag{11}$$

We introduce sequences $S_n$ containing the product of $n$ operators and sum over all possible choices of $S_n$. Since it is convenient for computer simulations if all sequences are of the same length, the sum over $n$ is truncated at a sufficiently large length $\mathcal{L}$, which is determined dynamically during the simulation. Sequences with $n < \mathcal{L}$ are padded to length $\mathcal{L}$ with trivial operators $\mathcal{H}_{0,0}$ [44, 45]. This procedure is justified by the observation that sequences longer than a certain $\mathcal{L} \sim \beta N$ only have an exponentially small contribution to the partition function, leading to an exponentially small truncation error [45, 55]. Taking into account a combinatorial factor considering the $\mathcal{L} - n$ inserted trivial operators, the partition function can be written as

$$Z = \sum_{\{|\alpha\rangle\}} \sum_{\{S_\mathcal{L}\}} \frac{\beta^n}{n!} \frac{n!(\mathcal{L}-n)!}{\mathcal{L}!} \langle\alpha| \prod_{l=1}^{\mathcal{L}} \mathcal{H}_{i(l),j(l)} |\alpha\rangle = \sum_{\omega\in\Omega} \pi(\beta, \omega). \tag{12}$$

We can define an SSE configuration space $\Omega = \{|\alpha\rangle\} \times \{S_\mathcal{L}\}$, consisting of the computational basis $\{|\alpha\rangle\}$ of the model and the set of sequences $\{S_\mathcal{L}\}$, which are an additional dimension that can be associated with a discretised imaginary time. The configuration space $\Omega$ is sampled using Markov chain Monte Carlo. The simulation starts with an initial configuration $\omega^0$ that is constantly updated. During the updates, the operator sequence $S_\mathcal{L}$ and the state $|\alpha\rangle$ are changed in accordance to their weight $\pi(\beta, \omega)$. A full Monte Carlo step consists of two types of updates, namely the diagonal and off-diagonal update which are alternately executed. In the diagonal update, the sequence $S_\mathcal{L}$ is traversed while propagating the state

$$|\alpha(p)\rangle = \prod_{l=1}^{p} \mathcal{H}_{i(l),j(l)}|\alpha\rangle. \tag{13}$$

In every step $p$, constant operators $\mathcal{H}_{i,i}$ and Ising operators $\mathcal{H}_{i,j}$ can be exchanged by trivial operators $\mathcal{H}_{0,0}$ and vice versa based on the Metropolis-Hastings algorithm [56, 57]. The proposal probability to insert a non-trivial operator $\mathcal{H}_{i,j}$ is given by

$$q(\mathcal{H}_{i,j}) = \frac{M_{ij}}{\sum_i h_i + 2\sum_{i\neq j}|J_{i,j}|}, \quad \text{with} \quad M_{ij} = \langle\alpha(p)|\mathcal{H}_{i,j}|\alpha(p)\rangle, \tag{14}$$

and is, in contrast to the pure nearest-neighbour TFIM, different for each operator. The proposal probability to insert a trivial operator is always 1. The probabilities to accept the exchange of an operator are given by

$$P(\mathcal{H}_{0,0} \to \mathcal{H}_{i,j}) = \min\left(1, \frac{\beta\left(\sum h_i + 2\sum |J_{i,j}|\right)}{\mathcal{L} - n}\right),\tag{15}$$

$$P(\mathcal{H}_{i,j} \to \mathcal{H}_{0,0}) = \min\left(1, \frac{\mathcal{L} - n + 1}{\beta\left(\sum h_i + 2\sum |J_{i,j}|\right)}\right).\tag{16}$$

Because only diagonal operators are exchanged, the sequence $S_{\mathcal{L}}$ and its weight can change during the diagonal update but the state $|\alpha\rangle$ remains the same. On the other hand, in the off-diagonal update, the sequence $S_{\mathcal{L}}$ as well as the state $|\alpha\rangle$ can change. By exchanging constant operators $\mathcal{H}_{i,i}$ with field operators $\mathcal{H}_{i,0}$ and vice versa, whole clusters of spins can be flipped without changing the weight $\pi(\beta, \omega)$ of the configuration. This is done via the quantum cluster update discussed in Ref. [44]. For an in-depth explanation on the setup of an SSE for the TFIM we recommend Refs. [13, 44, 45, 55].

## 3.1 Observables

Diagonal observables can be measured during the diagonal Monte Carlo updates. In the following, the Monte Carlo average is denoted by $\langle\ldots\rangle_{\mathrm{MC}}$ and the thermal average by $\langle\ldots\rangle$. The statistics of diagonal observables can be improved by measuring them at every intermediate state in the operator sequence

$$\langle \mathcal{O} \rangle = \frac{1}{Z} \sum_{\omega \in \Omega} \pi(\beta, \omega) \frac{1}{\mathcal{L}} \sum_{l=0}^{\mathcal{L}-1} \langle \alpha^{(l)} | \mathcal{O} | \alpha^{(l)} \rangle =: \left\langle \frac{1}{\mathcal{L}} \sum_{l=0}^{\mathcal{L}-1} \langle \alpha^{(l)} | \mathcal{O} | \alpha^{(l)} \rangle \right\rangle_{\mathrm{MC}},\tag{17}$$

because every cyclic permutation of the operator sequence is a valid configuration with the same weight $\pi(\beta, \omega)$ due to the cyclic property of the trace [45]. We focus mostly on the measurement of the order parameter, i.e. the moments of the $z$-magnetisation, since it is an easily accessible diagonal observable and can be used for various methods of data analysis. The squared magnetisation is given by

$$\langle m^2 \rangle = \left\langle \frac{1}{N^2}\Big(\sum_{i=1}^{N} \sigma_i^z\Big)^2 \right\rangle.\tag{18}$$

Besides diagonal observables at a single imaginary time, also imaginary-time integrated correlations can be calculated by summing over contributions from every step in the sequence $S_{\mathcal{L}}$. The squared imaginary-time integrated magnetisation is defined as

$$\langle m_{\mathrm{int}}^2 \rangle = \left\langle \left(\frac{1}{\beta}\int_0^\beta m(\tau)\, d\tau\right)^2 \right\rangle = \frac{1}{Z}\sum_{\omega \in \Omega}\pi(\beta, \omega)\frac{1}{\mathcal{L}^2}\left(\sum_{l=0}^{\mathcal{L}-1}\langle\alpha^{(l)}|\frac{1}{N}\sum_{i=1}^{N}\sigma_i^z|\alpha^{(l)}\rangle\right)^2.\tag{19}$$

We will use this quantity to determine the critical point via the imaginary-time integrated Binder ratio $V_{\mathrm{int}}$ and compare with the results of Ref [40] in App. C. Furthermore, also correlation functions can be calculated using SSE [45] (for a detailed derivation see Ref. [13]). In this work we want to investigate the distribution of the local susceptibility to access the dynamic scaling in the Griffiths phase (see Sec. 4.4). As a special case of the correlation function, this observable can be calculated by

$$\chi_{i,\mathrm{local}} = \int_0^\beta \langle\sigma_i^z(\tau)\sigma_i^z(0)\rangle\, d\tau = \left\langle \frac{\beta}{n(n+1)}\left(n + \Big(\sum_{p=0}^{n-1}\sigma_{i,p}^z\Big)^2\right)\right\rangle_{\mathrm{MC}},\tag{20}$$

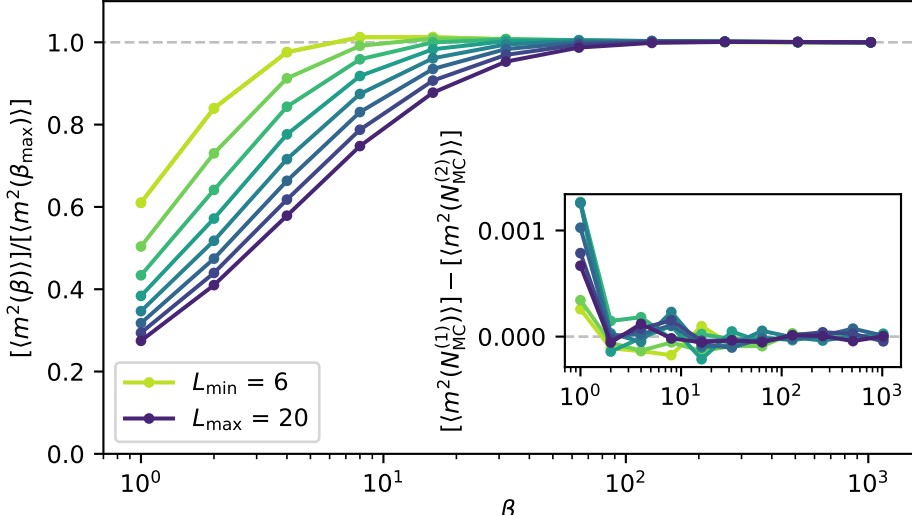

Figure 1: Temperature dependence of the averaged magnetisation at $h = h_c = 1$ for the $h$-box RTFI chain measured during the beta doubling procedure. As soon as the temperature is smaller than the finite energy gap, the magnetisation curves converge. In the inset the difference of the averaged magnetisation measured in the first and second set of $N_{MC}$ Monte Carlo steps is shown. Since there are only statistical deviations, the system is equilibrated.

where $\sigma^z_{i,p}$ is the spin state measured at propagation step $p$ in the sequence iterating only over the $n$ non-trivial operators.

## 3.2 Convergence to zero temperature

The SSE approach is a finite-system and finite-temperature QMC algorithm to sample thermal averages of observables. In order to study the quantum criticality of the RTFIM with this method, the simulation has to be performed at sufficiently low temperature for thermal fluctuations to be negligible. As a finite energy gap is ensured in any finite system close to the phase transition, one can, in principle, always find a temperature that is low enough to suppress excitations and effectively sample zero-temperature observables. An efficient and controlled way to cool the system down is the beta doubling method introduced by A. Sandvik [58] (see Refs. [10, 11, 13] for further adaptions). The inverse temperature $\beta = 1/T$ is doubled in every step until the desired temperature is reached. In every beta doubling step, the system is updated by two sets of $N_{MC}$ Monte Carlo steps, followed by a check whether the length of the sequence $\mathcal{L}$ is still sufficient. In the next $\beta$-step, we set the initial sequence to two copies of the sequence from the previous step glued together, since $\mathcal{L} \sim \beta$. If the system has been sufficiently cooled, observables like the squared magnetisation in Fig. 1 converge to a constant value. In order to observe the convergence, observables are measured in every step. Furthermore, equilibration can be checked by comparing the observables measured during the two sets within a doubling step [58] (see inset of Fig. 1). For conventional scaling, the dependency on the linear system size $L$ of the convergence can be estimated from the dynamic exponent $z$. However, in the case of activated scaling, the energy gap does no longer close with $L^{-z}$ but closes exponentially (see Eq. (6)). Furthermore, the energy gap varies significantly from disorder configuration to disorder configuration since the distribution of energy gaps decays algebraically (see Eq. (32)), causing different disorder configurations to converge in temperature at different $\beta$ values. Since controlling the temperature dependence and equilibration of

each disorder realisation separately is impractical, one has to be extremely careful about being converged in temperature. Besides checking the convergence of the squared magnetisation, we also evaluate the temperature dependence of all observables considering the previous temperature steps of the beta doubling and compare the finite-size scaling results within these last temperature steps. It turns out, that even if the magnetisation is converged in temperature, composed observables like Binder ratios and distributions of critical points are more sensitive to the finite temperature requiring even more cooling. Observables like a local susceptibility that have an explicit dependency on the temperature are even more affected by finite temperature and may not converge at all. We will discuss this issue on the basis of our results in Sec. 5.3. For the results shown in this work we empirically determined the required temperature for each model and had to cool down systems to up to $\beta = 2^{17}$. It turned out that the defining properties for the needed temperature are the linear length scale and the type of disorder we considered, e. g. $h$-box disorder seemed to give smaller energy gaps and required therefore a lower temperature than $h$-fix. Lower temperature comes with the drawback of longer simulation time and increased memory demand, since the length of the operator sequence $S_{\mathcal{L}}$ is proportional to $\beta$. However, it turned out that investigating a more extreme type of disorder results in faster convergence towards the IDFP critical exponents and is therefore worth the additional computational effort (see Sec. 5).

### 3.3 Disorder average and Sobol sequences

A crucial part of the evaluation of observables in systems with disorder is averaging. Since both the one- and two-dimensional RTFIM violate the Harris criterion, they are not self-averaging in the vicinity of the critical point [27, 41, 49], which is the region we are interested in. Therefore, it is not necessarily ensured that observables measured on large $L$ systems are averaging significantly faster in disorder configurations than small $L$ systems. In general, it is hard to determine a minimal amount of disorder configurations and to estimate meaningful errors on data points. To check if our observables are converged we use the bootstrapping method to estimate the variance of, e. g. intersections of Binder ratios.

Another degree of freedom is the choice of the number of Monte Carlo steps. For averaged observables, the general rule is that the number of Monte Carlo steps should be kept small in favor of more disorder realisations in the same time [58]. Even though an individual disorder realisation itself is poorly averaged in terms of Monte Carlo steps, the average over many disorder averages makes up for that by sampling different areas of the Monte Carlo configuration space. In practice, the lower bound for $N_{\mathrm{MC}}$ is reached when the system does no longer equilibrate [58]. This is checked during the beta doubling method (see inset of Fig. 1). Empirically, it turned out that $N_{\mathrm{MC}} < 100$ is sufficient for our models. Even for the evaluation of quantities that directly depend on sample-dependent observables (see Sec. 4.2), we did not increase $N_{\mathrm{MC}}$ since statistical inaccuracies on each sample-dependent observable vanish in the limit of many samples in a distribution.

The disorder configuration space for both $h$-fix and $h$-box is a high-dimensional space $I_s = [0,1]^s$, where $s$ is the number of random bonds $J_{i,j}$ or the number of random bonds and random fields $h_i$ respectively. In order to use the available computing time as efficiently as possible, it is important to already choose a representative set of samples in the disorder configuration space, i. e. more generally speaking an integral of a high-dimensional space,

$$\int_{I_s} f(u)\, du \approx \frac{1}{N} \sum_{i \in P} f(x_i),\tag{21}$$

should be approximated as good as possible by a set $P = \{x_1, \dots, x_n\}$ of samples $x_i \in I_s = [0,1]^s$. The common choice is to use a pseudo-random number generator. A possibility to sample

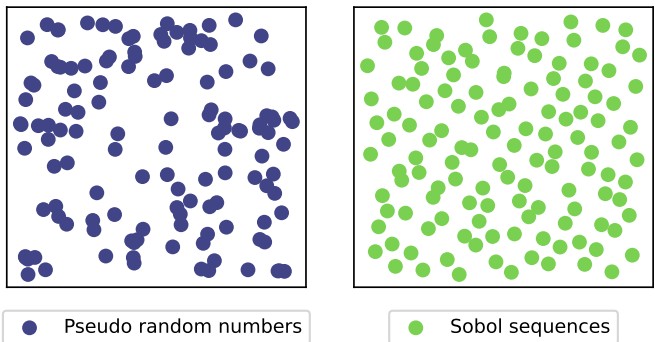

Figure 2: Example for a set of two-dimensional random numbers (left) and Sobol sequences (right). This would correspond to the disorder configuration space of a two-spin system with $h$-fix disorder. In general the dimension of the disorder configuration space is the number of bonds for $h$-fix disorder and the number of bonds plus the number of sites for $h$-box disorder.

the disorder configuration space more evenly (see Fig. 2) is to use quasi-random numbers instead [59–61]. Quasi-random numbers are sequences with a low discrepancy. The discrepancy of a set $P$,

$$D_n(P) = \sup_{B \in J} \left| \frac{(\#x_i \in B)}{n} - \lambda_s(B) \right|, \quad \text{with} \quad J = \left\{ \prod_{i=1}^{s} [a_i, b_i) \right\}, \tag{22}$$

is defined such, that the discrepancy of a set is smallest if the proportion of points $x_i$ in every subspace $B$ is equal to the measure of the subspace $\lambda_s(B)$. A set with a low discrepancy minimises holes in the $s$-dimensional space $I_s$ and all lower-dimensional faces of $I_s$. A typical choice for a low-discrepancy sequence that can be generated very efficiently by a computer and performs well for high-dimensional configurations spaces are Sobol sequences [62]. In App. A we elaborate on the advantage of the use of Sobol sequences instead of pseudo-random numbers. Note, that we do not use Sobol sequences for the Markov chain Monte Carlo sampling, but only for the sampling of the disorder configuration space.

## 4 Data analysis for disordered systems

A primary result of the RG study of quantum phase transitions in pure systems is finite-size scaling, which describes how certain observables behave in dependence of the system size and allows one to extract critical points and exponents from observables measured for different system sizes [28, 63–69]. For a pure system the general scaling form of an observable $\mathcal{O}$ in the vicinity of a quantum phase transition ($T = 0$) is given by

$$\mathcal{O}(r, L) = L^{-\omega/\nu} f_{\mathcal{O}}(r L^{1/\nu}), \tag{23}$$

where $r = (h - h_c)$ is the control parameter, $\omega$ the critical exponent of the observable with respect to the control parameter $r$ and $f_{\mathcal{O}}$ a scaling function. Eq. (23) holds for a pure system with the same shifting and rounding exponent (see Ref. [70] for an introduction to shifting and rounding exponents in pure systems). However, this scaling form describes only the leading behaviour. In addition, there are corrections to finite-size scaling [68] which are usually notoriously hard to fit. These corrections turn out to be strong in systems with disorder [35, 40, 51, 71]. In general, one obtains distributions of observables instead of individual

values. To be able to apply the finite-size scaling forms as described above, the disorder must be averaged at one point. At which step of the process exactly to average over disorder realisations is another degree of freedom, which can also have an influence on the scaling of an observable. In this work, we use several Binder ratios, a sample-replication method and the scaling of the averaged magnetisation to determine the position of the critical point and critical exponents $\nu$ and $\beta$. There are further methods in literature, that also provide estimates for the critical point and exponents which, however, turned out to be less precise for the systems we investigated. We will discuss these methods in App. C.

## 4.1 Intersections of Binder ratios

A common method to determine the position of the critical point are the intersections of Binder ratios $V$ [68]. These are defined by the ratio of the second and fourth moment of the magnetisation

$$V = \frac{1}{2}\left(3 - \frac{\langle m^4 \rangle}{\langle m^2 \rangle^2}\right).$$

(24)

For scalar order parameters, the ratio $\langle m^4 \rangle / \langle m^2 \rangle^2$ is modified by the constants presented above in order to have $V = 1$ in the ordered phase and $V = 0$ in the disordered phase. The importance of Binder ratios comes from the fact that

$$V(r, L) = f_V(r L^{1/\nu}),$$

(25)

is independent of $L$ at the critical point $r = 0$, since the scaling powers of the moments of the order parameter cancel. Therefore, the intersections of $V(L)$ can be used to determine the critical point. In the case of systems with quenched disorder, there are various ways to define the Binder ratio, depending on the stage at which the average over the disorder is taken [37, 39, 40, 71, 72]. Two reasonable choices are

$$V_1 = \frac{1}{2}\left(3 - \frac{\left[\langle m^4 \rangle\right]}{\left[\langle m^2 \rangle\right]^2}\right), \qquad V_2 = \frac{1}{2}\left[\left(3 - \frac{\langle m^4 \rangle}{\langle m^2 \rangle^2}\right)\right],$$

(26)

where $[\dots]$ denotes the disorder average. The stage at which the disorder average is taken is not just a numerical finesse, but the averaging has an essential influence on the subleading scaling behaviour (see Fig. 3). In App. C we will elaborate on more possible definitions for Binder-like ratios (see e.g. Ref. [72] and [73]) including imaginary-time integrated Binder ratios [39, 40, 74]. Since the corrections to finite-size scaling are very prominent in systems with disorder (see also Ref. [71]), the Binder ratios do not intersect almost perfectly at the critical point like for the pure system [68]. Instead, the region where the curves for different system sizes intersect is very broad (see Fig. 3). In order to determine the intersection points of the curves with high accuracy, very good data quality is required, i.e. averaging over a large number of disorder configurations. With increasing system size, the Binder ratios intersect shallower, which makes it even more difficult to determine the intersection point when there are statistical inaccuracies. We expect the intersections to scale with increasing system size towards the critical point. The functional dependence of the scaling is, to the best of our knowledge, not known. We assume a very general algebraic dependence, which is in agreement with our data shown in Sec. 5. There is, however, a theory for the scaling of intersections of Binder ratios for pure systems by Ref. [75], which we also applied to our data. Unfortunately, this does not seems to fully capture the corrections of the systems investigated in this work. We elaborate on this in App. C in more detail.

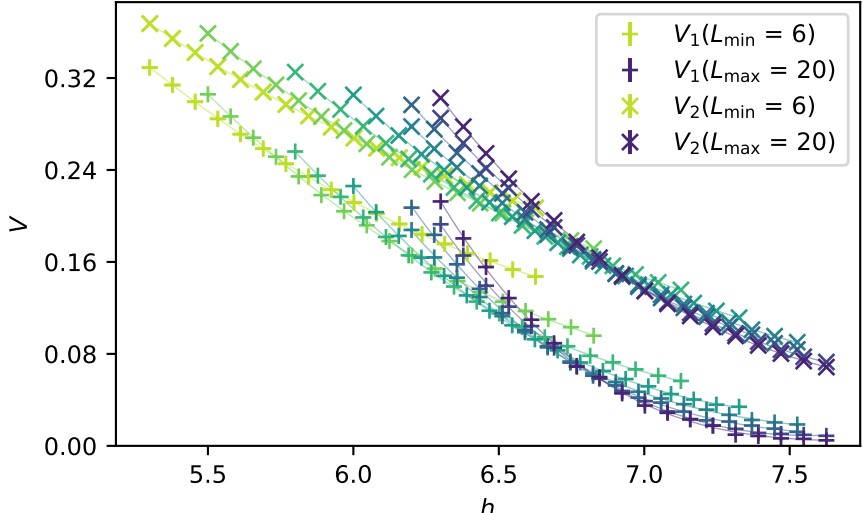

Figure 3: Binder ratios $V_1$ and $V_2$ for the $h$-box RTFIM on a square lattice using different linear system sizes $L$. Even though both Binder ratios are calculated from the same magnetisation curves, they intersect at different positions, i.e. are affected by different corrections.

## 4.2 Sample-replication method

A sample-replication method [35,52,76] is used to determine the position of the critical point for each realisation of the quenched disorder. The magnetisation $\langle m^2(h, L)\rangle$ of a certain system and also the magnetisation $\langle m^2(h, 2L)\rangle$ of the doubled system[1] are calculated. Then the ratio of the magnetisations,

$$\Phi(h, L) = \frac{\langle m^2(h, L)\rangle}{\langle m^2(h, 2L)\rangle}, \tag{27}$$

is calculated as a function of the control parameter $h$. $\Phi(h, L)$ is 1 in the ordered phase and 1/2 in the disordered phase. The pseudo-critical point is defined at the point where the ratio drops down (see Fig. 5 (left)). Since $\Phi(h, L)$ does not drop sharply, especially for small systems, the point of the drop is determined by fitting a Fermi-distribution-like function,

$$f(\tilde{r}, a) = \frac{1}{2}\left(\frac{1}{1 + \exp(a\tilde{r})} + 1\right), \tag{28}$$

to $\Phi(h, L)$. The control parameter is $\tilde{r} = \log(h) - \log(\tilde{h}_c)$, where $\tilde{h}_c$ is the sample-dependent pseudo-critical point and $a$ is a free parameter depending on the "rounding" of the curves. Eq. (28) is an empirical choice that turned out to be suitable since it captures the step-function like behaviour for large system sizes ($a \to \infty$) as well as the rounded curves of smaller system sizes. For this method we use the logarithmic control parameter $\tilde{r}$ instead of $h - \tilde{h}_c$. In the thermodynamic limit, this choice does not play a role for scaling, but for small system sizes this choice has proven to be advantageous. One motivation for this is that for the one-dimensional RTFIM it is known that the distribution of pseudo-critical points is approximately a log-normal distribution (see Eq. (4)). For the two-dimensional system, there are no analytical results for the form of the distribution. However, in numerical studies such as [35] or this work, the

---

[1]We define a doubled system as a system, where we connect two copies of the original system as depicted in Fig. 4.

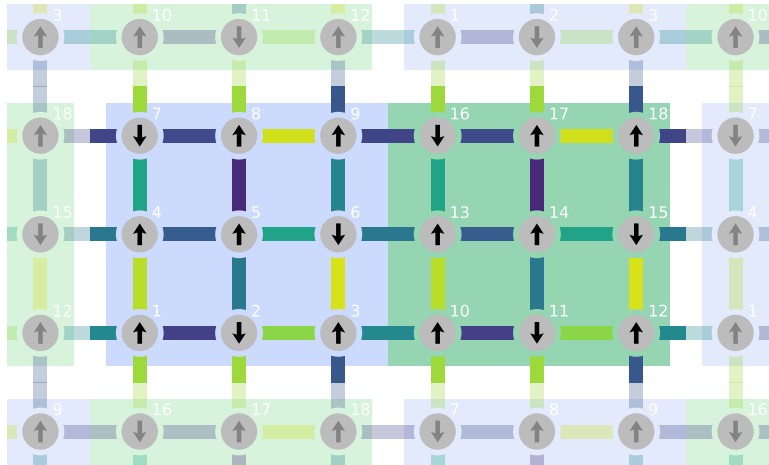

Figure 4: The doubled system consists of two copies of the original system which are periodically connected. In one dimension the periodic boundary conditions for a doubled system are trivial. The periodic coupling of the doubled system in two dimensions resulting in an isotropic lattice with alternating pattern of the original and copied system is shown here.

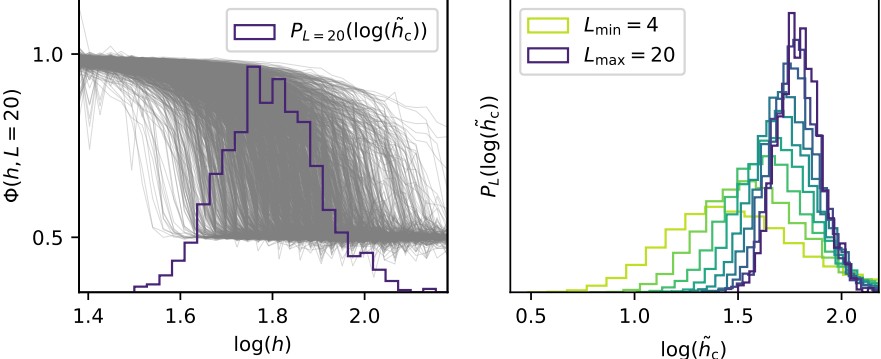

Figure 5: Left: The ratio $\Phi(h, L = 20)$ is shown for many disorder realisations (grey curves) for the $h$-box RTFIM on the square lattice. Every curve drops at its sample-dependent critical point from 1 to 0.5. The distribution of pseudo-critical points is obtained by determining the position of the drop-off for each curve. Right: The distribution of pseudo-critical points is shown for different system sizes.

distribution also looks similar to a log-normal or normal distribution (see Fig. 5 (right)). From the obtained distributions of pseudo-critical points, critical shift and width exponents $\nu_s$ and $\nu_w$ and the critical point $h_c$ can be extracted by finite-size scaling using the mean and the standard deviation [52]

$$\left[\log(\tilde{h}_c(L))\right]_{P_L(\log(\tilde{h}_c))} = \log(h_c) + A \cdot L^{-1/\nu_s}, \qquad \sigma(P_L(\log(\tilde{h}_c))) = B \cdot L^{-1/\nu_w}. \tag{29}$$

## 4.3 Scaling of the average magnetisation

A common finite-size scaling technique to extract or verify critical exponents and critical points is a data collapse (see e. g. Ref. [10, 11]). This method rescales observables of different system sizes, e. g. magnetisation curves, with respect to both their amplitude and position. Both shifting and rounding can be tackled by this approach and all rescaled curves should lie on top of each other in the relevant scaling window [70] (see Eq. (23)). However, it turned out, that this method performs poorly for the RTFIM using small system sizes (see App. C). Since corrections to scaling are strong, it seems a single fit cannot capture the right exponents using this method. In order to keep the influence of corrections to scaling as low as possible, we consider the scaling of the magnetisation directly at the critical point $r = 0$, i. e. we only fit the critical exponent of the observable

$$\left[\langle m^2(r=0,L)\rangle\right] \sim L^{-2\beta/\nu_{av}}. \tag{30}$$

On the other hand, we also want to fit the average correlation length exponent $\nu_{av}$. Therefore, we expand the scaling form of the magnetisation (see Eq. (23)) close to the critical point and consider the observable

$$1 - \frac{\left[\langle m^2(r=\delta)\rangle\right]}{\left[\langle m^2(r=0)\rangle\right]} = 1 - \frac{f(0) + \frac{\partial f}{\partial \delta}|_{\delta=0}\delta L^{1/\nu_{av}} + O(\delta^2)}{f(0)} \sim L^{-1/\nu_{av}}, \tag{31}$$

where the exponent $\omega/\nu_{av} = 2\beta/\nu_{av}$ cancels out in first order and we can therefore only extract the average correlation length exponent $\nu_{av}$. The separate consideration of the two finite-size effects has proven to be more fruitful in this work, as corrections are now dealt with individually.

## 4.4 Dynamic scaling

Because of the strong influence of rare regions, especially in the disordered phase, the distribution of energy gaps,

$$P(\Delta) \sim \Delta^{\frac{d}{z'}-1}, \tag{32}$$

has an algebraic tail for small energies $\Delta$ [50]. This follows directly from the exponential relation between typical length scales and energy gaps (see Eq. (6)). Since the exponent relates energy and length scales, it is denoted by $z'$, even though $z$ is not defined for critical points governed by the IDFP. Instead of calculating the energy gap directly, we determine the local susceptibility as described in Eq. (20). The local susceptibility can be rewritten in terms of energy gaps by evaluating the integral over imaginary time:

$$\chi_{i,\text{local}} = \int_0^\beta \langle \sigma_i^z(\tau)\sigma_i^z(0)\rangle \, d\tau = \int_0^\beta \frac{1}{Z} \sum_{|\alpha\rangle,|\gamma\rangle} \langle\alpha|e^{\mathcal{H}(\tau-\beta)}\sigma_i^z e^{-\mathcal{H}\tau}|\gamma\rangle\langle\gamma|\sigma_i^z|\alpha\rangle \, d\tau \tag{33}$$

$$= \frac{1}{Z} \sum_{|\alpha\rangle\neq|\gamma\rangle} \frac{1}{E_\gamma - E_\alpha}\left(e^{-\beta E_\alpha} - e^{-\beta E_\gamma}\right)|\langle\alpha|\sigma_i^z|\gamma\rangle|^2 \tag{34}$$

$$= \frac{2}{Z} \sum_{|\alpha\rangle\neq|\gamma\rangle} \frac{|\langle\alpha|\sigma_i^z|\gamma\rangle|^2}{E_\gamma - E_\alpha}e^{-\beta E_\alpha} \sim \frac{1}{\Delta}, \tag{35}$$

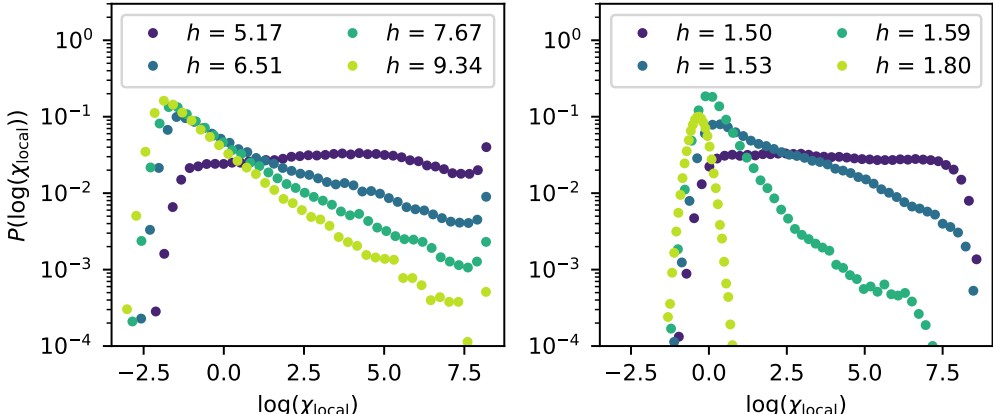

Figure 6: Distribution of the local susceptibility for the $h$-box (left) and $h$-fix (right) RTFIM on the square lattice. The distributions show an exponential tail in the Griffiths phase. From the slope of the exponential tail, the exponent $z'$ is determined. Outside the Griffiths phase (right figure for $h = 1.80$), the distribution has a Gaussian form.

which is for small temperatures indirectly proportional to the energy gap $\Delta = E_1 - E_0$. Analogous to the exponential tail of small energy gaps, we expect to observe an exponential tail in the limit of large local susceptibilities. In Fig. 6 you can see the exponential tail of the distribution of the local susceptibility in the Griffiths phase. Outside the Griffiths phase the distribution has the form of a normal distribution. This behaviour can be used to roughly determine the phase boundary between the Griffiths phase and the paramagnetic phase. Considering the distribution of $\log(\chi_{\mathrm{local}})$, we get access to the exponent $z'$ [38, 39, 51]

$$\log\big[P(\log\chi_{i,\mathrm{local}})\big] \sim -\frac{d}{z'}\log\chi_{i,\mathrm{local}}, \tag{36}$$

i.e. the dynamic scaling in the Griffiths phase. The point where $z'$ diverges can be used to determine $h_{\mathrm{c}}$ and the point where dynamic scaling disappears can be used to determine the phase boundary of the Griffiths phase (if it exists). However, finite temperature plays a huge role for this method, as we will point out in Sec. 5.3.

# 5 Results

This section is structured as follows: The finite-size scaling methods introduced in the previous section are now applied to both the one- and two-dimensional RTFIM. First, we use the intersections of Binder ratios to accurately determine the critical point $h_{\mathrm{c}}$. Then the sample-replication method is used to validate the position of the critical point as well as to determine the shift and width exponents $\nu_{\mathrm{s}}$ and $\nu_{\mathrm{w}}$. To extract the critical exponents $\beta$ and $\nu_{\mathrm{av}}$, we consider the averaged magnetisation at the critical point, whose position we determined from the previous methods. We first show our results for the one-dimensional RTFIM in Sec. 5.1, which we can compare with the exact RG results [28], several numerical studies [34, 37, 52, 71] and results using the Jordan-Wigner method described in App. B. In Sec. 5.2 we turn to the two-dimensional model applying the same methods as in one dimension and furthermore investigate the dynamic scaling via the local susceptibility. Finally, in Sec. 5.3, we want to stress the importance of temperature effects. The raw data used for the presented results as well as processed data are provided in Ref. [77].

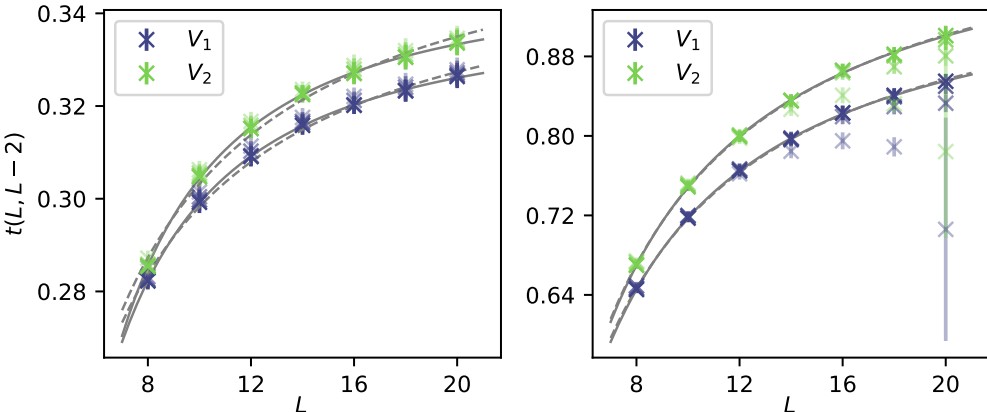

Figure 7: Intersections of neighboring Binder ratios ($L$ and $L-2$) are determined for $h$-fix (left) and $h$-box (right) disorder on the one-dimensional RTFIM. Both definitions of the Binder ratio $V_1$ and $V_2$ do not intersect at a single point, but are influenced by corrections to scaling. The faded data points are the intersection points from simulation with higher temperature ($\times 2$, $\times 4$ and $\times 8$ with decreasing saturation) measured during the beta doubling procedure. The solid and dashed lines are algebraic and $1/L$ fits to the data points respectively.

## 5.1 One-dimensional chain

For the RTFIM on the linear chain we considered system sizes in the range $L = 6$ to $L = 20$. For better comparison we simulate the same linear system sizes that we can achieve in two dimensions with sufficient convergence in temperature and number of disorder realisations. A verification for the validity of our methods and our results including data for larger system sizes is given in App. A. We reach temperature convergence using the beta doubling method introduced in Sec. 3.2 with $\beta_{\max} = 2^{12}$ for $h$-fix disorder and $\beta_{\max} = 2^{10}$ for $h$-box disorder. For the sample-replication method, where we also have to simulate the same system with twice the linear extent, we have to cool down the system even more: $\beta_{\max} = 2^{15}$ for $h$-fix disorder and $\beta_{\max} = 2^{17}$ for $h$-box disorder. We see, that the convergence in temperature is not a linear process for the RTFIM, but an exponential one. The finite-size gap determining our minimal temperature is dominated by activated scaling, which seems to be more extreme for $h$-box than $h$-fix. For the sample-replication method at least 3800 disorder realisations have been computed for each system size. For the Binder techniques and scaling of the magnetisation, where higher accuracy is necessary, at least 160000 disorder realisations have been computed for each system size. Furthermore, for the latter, different disorder realisations are taken for every point in the grid of the control parameter $h$.

In Fig. 7 the intersections $t(L, L-2)$ of the Binder ratios $V_1$ and $V_2$ are depicted for $h$-fix and $h$-box disorder. In contrast to pure systems, the Binder ratios do not intersect at almost a single point but are shifted with their system size $L$. This shifting has also been investigated in Ref. [71] for the antiferromagnetic RTFI chain. However, it is not clear with which functional dependency the intersections scale. The scaling of the intersections cannot be explained using the standard scaling function of the magnetisation but originates presumably from corrections to scaling. We elaborate on the possible form of these corrections in App. C applying the framework described in Ref. [75]. In Ref. [71], the scaling of the Binder intersections was assumed to be $1/L$, which also fits our data quite well (see dashed lines in Fig. 7). However, with regard to the results of the two-dimensional system, where this seems to be no longer given, we decided to choose the fit more general as an algebraic fit. The fits are shown in

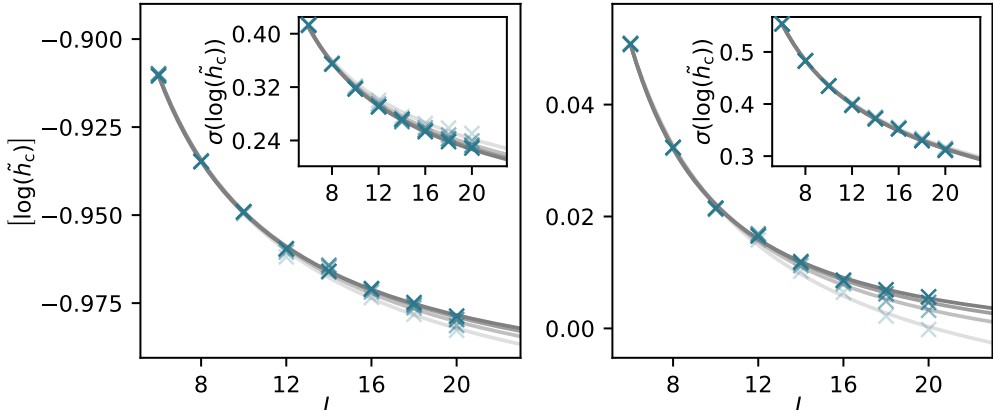

Figure 8: Sample-replication method applied to the linear chain with $h$-fix (left) and $h$-box (right) disorder. From algebraic fits to the curves $h_c$ and the exponents $\nu_s$ and $\nu_w$ are extracted. The faded data points are the pseudo-critical points from simulation with higher temperature ($\times 2$, $\times 4$ and $\times 8$ with decreasing saturation) measured during the beta doubling procedure.

Fig. 7 (solid lines) and result in critical points $h_c(V_1) = 0.340$ and $h_c(V_2) = 0.345$ for $h$-fix and $h_c(V_1) = 0.975$ and $h_c(V_2) = 1.038$ for $h$-box. This is fairly close to the literature values $h_c = 1/e \approx 0.368$ and $h_c = 1$ [48] considering the very small system sizes we computed. The varying results for the two definitions $V_{1/2}$ of the Binder ratio show the relevance of the freedom at which point the average over an observable is taken. There are more possible choices to define the Binder ratio mentioned in App. C, which led to critical points that are less in line with the literature values. In Fig. 7 one can also see the influence of the finite temperature, as we have calculated the intersection points for each of the last four temperature steps in the beta doubling. It turned out that the finite temperature still has a strong impact on the Binder ratios and their intersections, although the averaged magnetisation curves are already converged.

In Fig. 8 one can see the mean and standard deviation of the distribution of sample dependent pseudo-critical points for the $h$-fix and $h$-box RTFI chain. The mean value is expected to scale towards the respective critical point with a shifting exponent $\nu_s$, the standard deviation is expected to decay with a width exponent $\nu_w$. Both exponents and the critical point were extracted using algebraic fits and are shown in Tab. 1. Considering Eq. (4) the same quantity can be calculated by looking at the distribution of

$$\log(\tilde{h}_c(L)) = \frac{1}{L}\left(\log\left(\prod_i |J_{i,i+1}|\right) - \log\left(\prod_i h_i\right)\right) , \qquad (37)$$

leading to a width exponent $\nu_w = 2$. However, $\left[\log(\tilde{h}_c(L))\right]$ using Eq. (37) does not depend on $L$, since we average over a sum of independent random numbers. We attribute the $L$-

Table 1: Critical points and exponents extracted from the distribution of pseudo-critical points of the RTFI chain compared with literature values.

|  | literature | $h$-fix | $h$-box |
|---|---|---|---|
| $h_c$ | $1/e \approx 0.368$ ($h$-fix), $1$ ($h$-box) [48] | 0.365 | 0.996 |
| $\nu_s$ | 1 [52] | 0.995 | 0.692 |
| $\nu_w$ | 2 [52] | 2.023 | 2.118 |

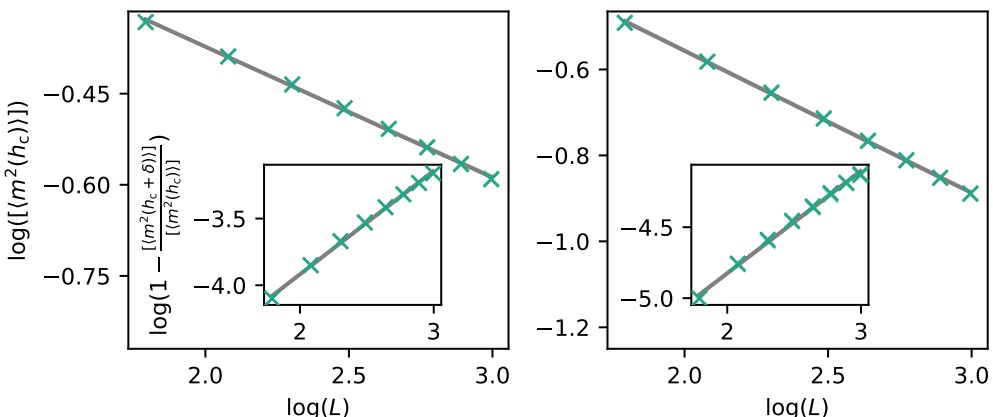

Figure 9: Using a linear fit, we can extract the exponent $\beta/\nu_{\mathrm{av}}$ from the averaged magnetisation at the critical point $h = h_{\mathrm{c}}$ for the $h$-fix (left) and $h$-box (right) RTFI chain. Furthermore, we use the observable defined in Eq. (31) to determine the critical exponent $\nu_{\mathrm{av}}$ from the same data. The faded data points are from the same simulation at higher temperature ($\times 2$, $\times 4$ and $\times 8$ with decreasing saturation) measured during the beta doubling procedure.

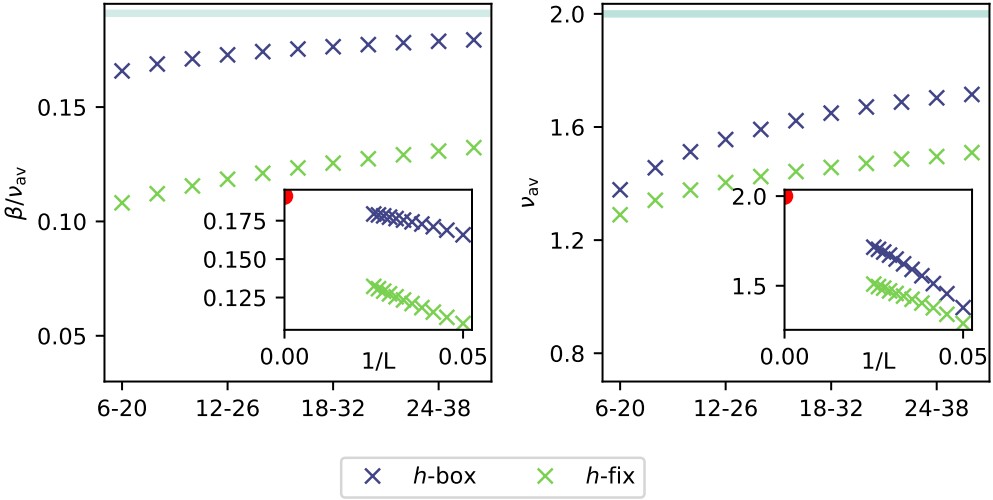

Figure 10: Critical exponents $\beta/\nu_{\mathrm{av}}$ and $\nu_{\mathrm{av}}$ of the RTFI chain extracted from linear fits to $\log\left(\left[\langle m^2(h_{\mathrm{c}})\rangle\right]\right)$ and the derivative of the magnetisation defined in Eq. (31). Here, data for larger system sizes generated by the Jordan-Wigner method described in App. B is used. We use successively larger system sizes to see the $L$-dependency of the critical exponents. In the inset the same data points are presented on $1/L$ scale, where $L$ denotes the largest system size considered for each data point.

dependency of $\left[\log(\tilde{h}_{\mathrm{c}}(L))\right]$ in our data to the neglected boundary term in the derivation of Eq. (4) [48]. The same shifting has been found in Ref. [52] using a different choice for the Ising coupling strengths $J_{i,i+1}$. Our finding of a very subtle shift for the $h$-box model is in agreement with the results of Ref. [34] observing almost no shift at all for the same model. The exponents using the model with $h$-fix disorder seem to agree better with the literature than the $h$-box model. However, we are cautious in interpreting these results to imply that the $h$-fix model converges faster to the correct exponents. One reason is the more extreme disorder of the $h$-box model leading to more problems regarding convergence in temperature and number of disorder realisations. In addition, in the case of irrelevant disorder, one would expect the shifting exponent $\nu_{\mathrm{s}}$ to continue to correspond to the clean exponent $\nu = 1$ and the width exponent $\nu_{\mathrm{w}}$ to correspond to the central limit theorem (CLT), i.e. $\nu_{\mathrm{w}} = 2/d = 2$ [27,52]. This means that in one dimension it is difficult to distinguish whether one has converged against the exponents of the IDFP or is still dominated by the pure/CLT exponents. For the two-dimensional square lattice, where $\nu_{\mathrm{s}}$ and $\nu_{\mathrm{w}}$ are expected to be different from the pure/CLT exponents, we can see that $\nu_{\mathrm{w}}$ of the $h$-fix model converges only very slowly from the CLT exponent to the IDFP exponent.

Given the averaged magnetisation at the critical point $\left[\langle m^2(h = h_{\mathrm{c}})\rangle\right]$ in a double logarithmic plot over the system size $L$ (see Fig. 9), we can extract the exponent $\beta/\nu_{\mathrm{av}}$ through a linear fit. We use the derivative of the magnetisation defined in Eq. (31) to extract the exponent $\nu_{\mathrm{av}}$ in the same way (see insets of Fig. 9). Although the last four temperature steps are again displayed, they are hardly visible. Compared to the intersections of Binder ratios, temperature convergence is much easier to achieve in this case. The extracted exponents of the magnetisation $\beta/\nu_{\mathrm{av}} = 0.108$ for $h$-fix and $\beta/\nu_{\mathrm{av}} = 0.165$ for $h$-box deviate from the literature value $(3-\sqrt{5})/4 \approx 0.191$. Also the exponent $\nu_{\mathrm{av}} = 2$ is not well met with our extracted values $\nu_{\mathrm{av}} = 1.290$ for $h$-fix and $\nu_{\mathrm{av}} = 1.384$ for $h$-box. It is noticeable that the exponents using $h$-box are closer to the literature values than for $h$-fix. This is in agreement with our results in two dimensions and the results of Ref. [35] and strengthens our hypothesis that models with more extreme disorder converge faster towards the IDFP exponents. Looking carefully at Fig. 9, one can see that the linear fit does not perfectly suit the behaviour of the data points. A deviation from linear behaviour indicates that corrections to the scaling form are present. We want to assess whether these corrections become less when we increase the system size. Therefore the magnetisation is computed for larger system sizes and with higher precision using the Jordan-Wigner method described in App. B. In Fig. 10 we compute the exponents again for successively larger sets of system sizes and observe an $L$-dependency towards the analytically known critical exponents. We see that the convergence is a lot slower for $h$-fix than for $h$-box. In the inset of Fig. 10 the same exponents are presented on an $1/L$ scale to demonstrate that the exponents seem to converge towards the correct exponent.

## 5.2 Two-dimensional square lattice

As for the one-dimensional case, we considered linear system sizes in the range $L = 6$ to $L = 20$ also for the two-dimensional RTFIM on the square lattice. Based on the experience we have gained from the one-dimensional model and the results from Ref. [35], we expect that the model with $h$-box disorder converges faster in system size $L$ towards the right exponents. Therefore we invested more computational effort for the study of the $h$-box than the $h$-fix model. For the sample-replication method, for which we simulate up to 800 spins, we have to cool down to $\beta_{\mathrm{max}} = 2^{15}$ for the $h$-box model and $\beta_{\mathrm{max}} = 2^{12}$ for the $h$-fix model. Here, at least 1700 disorder realisations have been computed for each system size. Considering the $h$-box model, for the averaged magnetisation, at minimum 11000 disorder realisations at $\beta_{\mathrm{max}} = 2^{11}$ have been calculated for each system size. For the Binder techniques at least 31000 disorder realisations per system size at $\beta_{\mathrm{max}} = 2^{12}$ have been computed. Again, different dis-

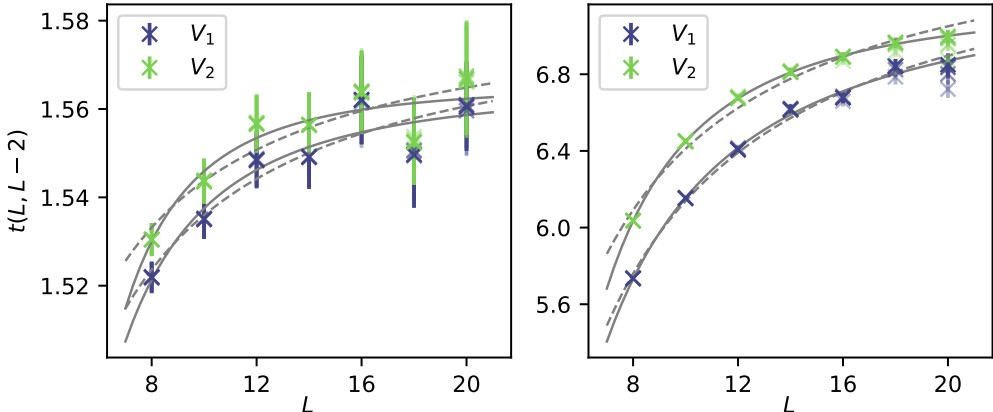

Figure 11: Intersections of neighbouring Binder ratios ($L$ with $L-2$) are determined for $h$-fix (left) and $h$-box (right) disorder on the square lattice RTFIM. As in the one-dimensional case, also for the two-dimensional system the Binder ratio $V_1$ and $V_2$ do not intersect at a single point, but are impacted by corrections to scaling. The faded data points are the intersection points from simulation with higher temperature ($\times 2$, $\times 4$ and $\times 8$ with decreasing saturation) measured during the beta doubling procedure. The solid and dashed lines are algebraic and $1/L$ fits to the data points respectively.

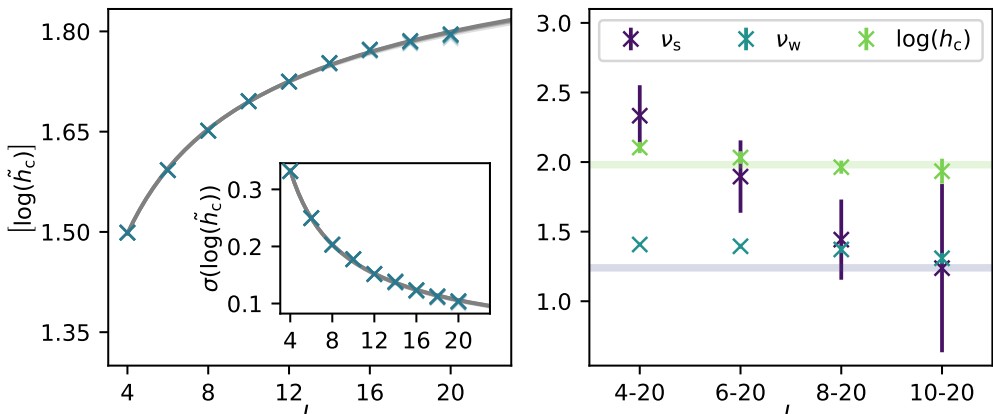

Figure 12: Sample-replication method applied to the RTFIM on the square lattice with $h$-box disorder. Left: The critical point $h_c$ and the exponents $\nu_s$ and $\nu_w$ are extracted from algebraic fits to the curves. The faded data points are the pseudo-critical points from simulation with higher temperature ($\times 2$, $\times 4$ and $\times 8$ with decreasing saturation) measured during the beta doubling procedure. Right: To observe the $L$-dependency of the critical point and exponents, smaller system sizes are stepwise excluded and fits are performed for the remaining data points. The solid lines indicate the critical point determined using the intersections of Binder ratios and the exponents $\nu_s$ and $\nu_w$ from Ref. [35].

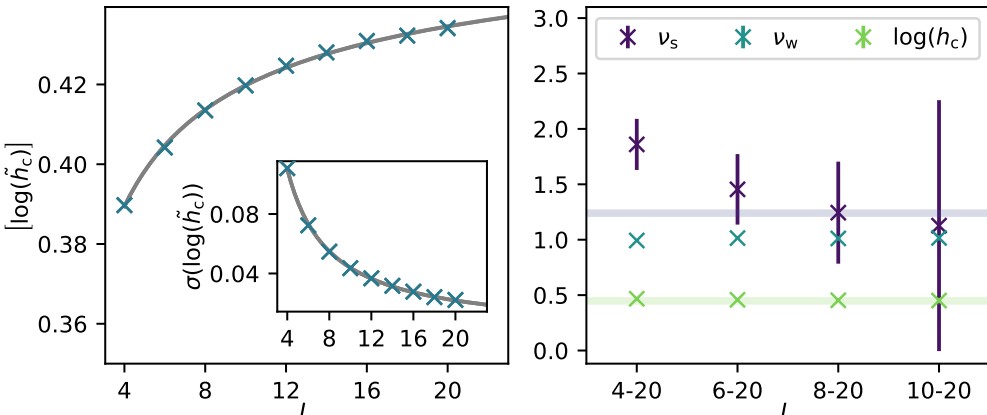

Figure 13: Equivalent figure to Fig. 12 for the RTFIM on the square lattice with $h$-fix disorder. Left: Algebraic fits to the mean and standard deviation of $P_L(\log(\tilde{h}_c))$. Right: $L$-dependency of the critical point and exponents. The solid lines indicate the critical point determined using the intersections of Binder ratios and the exponents $\nu_s$ and $\nu_w$ from Ref. [35].

order realisations are taken for every point in the grid of the control parameter $h$. In the case of the $h$-fix model we were able to reuse the data, that we computed for the sample-replication method.

In Fig. 11 we apply the method described for the one-dimensional chain to the Binder ratios $V_1$ and $V_2$ of the two-dimensional model. Here, the assumption of an $1/L$ scaling of the intersections is no longer in line with the data (see dashed lines in Fig. 11). Fitting an algebraic function to the data (solid lines in Fig. 11), we get $h_c(V_1) = 1.5653$ and $h_c(V_2) = 1.5654$ for the $h$-fix model and $h_c(V_1) = 7.300$ and $h_c(V_2) = 7.184$ for the $h$-box model. This is in agreement with a recent quantum Monte Carlo study [40] but neither with SDRG studies [35, 36, 53, 54] nor with the earlier Monte Carlo study [39]. The deviation from the SDRG results can be explained since the SDRG method can predict universal quantities like critical exponents very precisely but does not necessarily predict the "true" critical point as microscopic details are lost in the renormalisation process. The deviation from the earlier Monte Carlo study may be explained by temperature effects (see Sec. 5.3). In App. C we address the imaginary-time integrated binder ratios that were used in Ref. [40] to determine the critical point.

In Fig. 12 the results for the sample-replication method applied to the $h$-box model on the square lattice are shown. Algebraic fits lead to $h_c = 8.197$, $\nu_s = 2.33$ and $\nu_w = 1.408$. However, as also pointed out in Ref. [35], both exponents are still $L$-dependent, because of corrections to scaling. To assess the $L$-dependency, we stepwise exclude the smallest system and perform the fits again (see Fig. 12 (right)). Using this procedure, we see that the exponents seem to converge to the exponents determined in several RG studies [35, 36, 78], although the error bars calculated using the bootstrapping technique are quite large. The direction of the convergence, i.e. overestimating the exponents for small system sizes, is also consistent with the results from Ref. [35]. Furthermore, also the critical point shifts towards the value we extracted using the intersections of Binder ratios (see Fig. 12 (right)). $P_L(\log(\tilde{h}_c))$ for $h$-fix disorder is evaluated in Fig. 13. One can see that the convergence of $\nu_w$ towards the expected value is slower or even hardly visible compared to the $h$-box model being consistent with Ref. [35]. It seems like $\nu_w$ for the $h$-fix model is dominated by the central limit theorem, i.e. $\nu_w = 2/d = 1$. The exponent $\nu_s$ is affected by a large error, while the critical point coincides well with our estimate using the Binder techniques and the value predicted by Ref. [40].

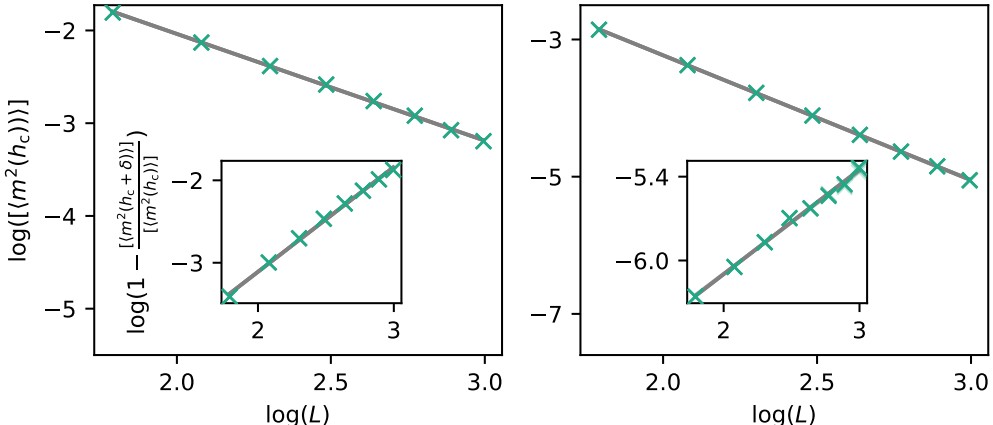

Figure 14: Using a linear fit, we can extract the exponent $\beta/\nu_{\mathrm{av}}$ from the averaged magnetisation at the critical point $h_c \approx 1.565$ and $h = h_c \approx 7.25$ obtained from the Binder methods for the $h$-fix (left) and $h$-box (right) RTFIM on the square lattice. Furthermore, we use the observable defined in Eq. (31) to determine the critical exponent $\nu_{\mathrm{av}}$ from the same data. The faded data points are from the same simulation at higher temperature ($\times 2$, $\times 4$ and $\times 8$ with decreasing saturation) measured during the beta doubling procedure.

In Fig. 14 the averaged magnetisation at the critical point $\left[\langle m^2(h = h_c)\rangle\right]$ is scaled for the two-dimensional RTFIM. On the basis of the results extracted for the intersection of Binder ratios, we assume the critical points to be $h_c \approx 7.25$ for the $h$-box model and $h_c \approx 1.565$ for the $h$-fix model. The extracted critical exponents for the $h$-box model are shown in Tab. 2 and compared with literature values. Both our values for $\beta/\nu_{\mathrm{av}}$ and $\nu_{\mathrm{av}}$ for the $h$-box model are compatible with the results of the SDRG and QMC studies available to far (see Tab. 2). Like $\nu_{\mathrm{s}}$ and $\nu_{\mathrm{w}}$ before, also the critical exponent $\nu_{\mathrm{av}}$ seems to be slightly overestimated compared to the SDRG results being in agreement with the sample-replication method. For the $h$-fix model we receive $\beta/\nu_{\mathrm{av}} = 0.578$ and $\nu_{\mathrm{av}} = 0.787$.

By successively removing smaller system sizes from the fit (see Fig. 15) we see, however, a strong $L$-dependence in the $h$-fix exponents. In contrast, the exponents for the $h$-box model seem to be less affected by finite-size corrections. Analogous to the results in one dimension (compare Fig. 10), the hypothesis arises that there is a finite-size crossover for weak disorder like the $h$-fix model, in which the exponents change from the ones of the pure system to the IDFP exponents of the disordered system. However, only looking at comparably small system sizes, we cannot rule out that the exponents converge towards different values as suggested by Choi et al [40]. We observe that the critical exponents extracted by scaling the averaged magnetisations are not completely stable under variation of the critical point but change slightly in a continuous way. Inaccuracies in the assumed location of the critical points lead to deviations in the critical exponents. Since the different methods we used to predict the critical point

Table 2: Critical exponents extracted from the averaged magnetisation of the $h$-box RTFIM on the square lattice by finite-size scaling compared with literature values, mostly obtained by SDRG using significantly larger system sizes.

| | [32] | [54] | [79] | [35] | [36] | [78] | [40] | this work |
|---|---|---|---|---|---|---|---|---|
| $\beta/\nu_{\mathrm{av}}$ | 1.0 | 1.01 | 0.96 | 0.982 | | | 0.94 | 0.909 |
| $\nu_{\mathrm{av}}$ | 1.07 | | 1.2 | 1.24 | 1.3 | 1.2 | 1.6 | 1.335 |

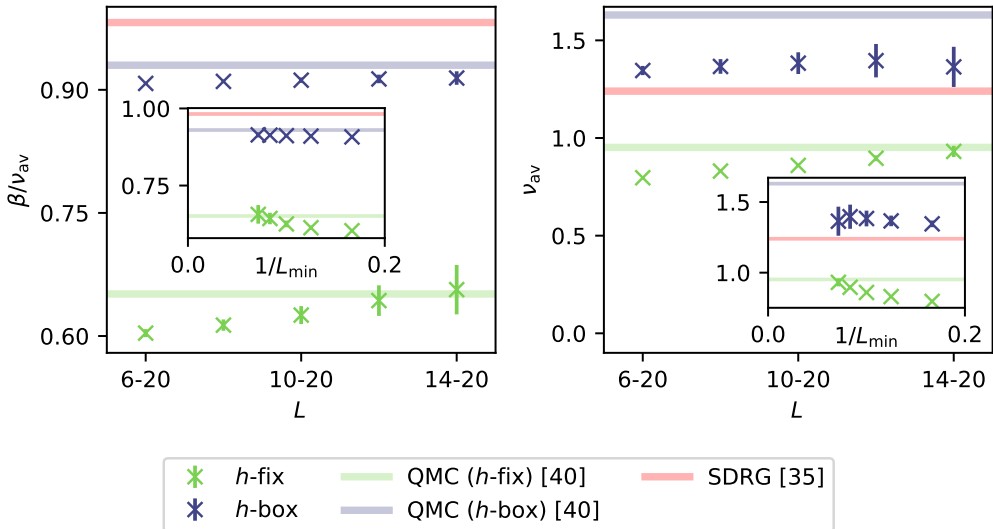

Figure 15: Critical exponents $\beta/\nu_{av}$ and $\nu_{av}$ of the $h$-fix and $h$-box RTFIM on the square lattice extracted from linear fits to $\log(\left[\langle m^2(h_c)\rangle\right])$ and the derivative of the magnetisation defined in Eq. (31). Smaller system sizes are successively excluded to see the $L$-dependency of the critical exponents. For comparison the results of Ref. [40] and [35] are depicted by solid lines. In the inset the same data points are presented on $1/L_{min}$ scale, where $L_{min}$ denotes the smallest system size considered for each data point.

determine the same value within reasonable accuracy, we are confident that also the critical exponents are not affected by a large error. As in the one-dimensional case, convergence in the temperature does not seem to be a problem either for this method (see faded data points in Fig. 14).

From the distribution of the local susceptibility (see e. g. Fig. 6), the exponent $d/z'$ is extracted from the linear slope of the exponential tail when double-logarithmically displayed [38,39,51]. The ratio $d/z'$ is shown in Fig. 16 for both the $h$-box and $h$-fix RTFIM on the square lattice. For both disorder types, the exponent approaches 0 at a certain point, corresponding to activated scaling, i.e. $z = \infty$ at the critical point. The estimate of the critical point gained from the condition $d/z' = 0$ is prone to both finite-size and temperature effects (see Sec. 5.3). In one dimension for both $h$-box and $h$-fix (not shown here, see e. g. Ref. [51]), the convergence looks similar to the two-dimensional $h$-box model. The two-dimensional $h$-fix model shows a different type of convergence (compare Fig. 16). In a recent QMC study [40], it was pointed out that there may be a fundamental difference between the $h$-box and $h$-fix model since the critical exponents of the latter were rather connected to the 2D transverse-field Ising spin glass universality class than the universality class of the IDFP. However, the presence of the exponential tail in the local susceptibility (see Fig. 6 (right)) hints towards an IDFP. It is striking that $d/z'$ in Fig. 16 (left) does not converge to zero at the critical point, as we expect from the previous analysis, but rather to the point predicted by Refs. [38,39] using the same technique. The reason for the inconsistency in the location of the critical point is not only a lack of extrapolation in $L$ in Fig. 16, but also the strong temperature effects, analysed in the following section.

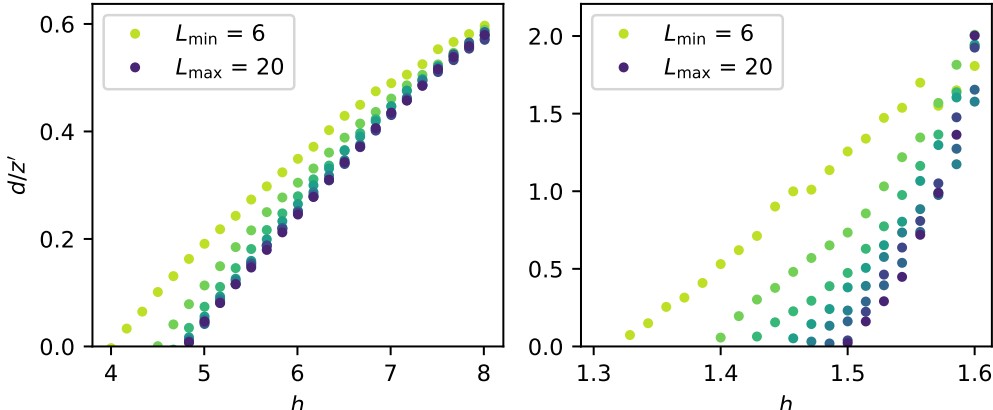

Figure 16: Exponent $d/z'$ extracted from the distribution of the local susceptibility for the $h$-box (left) and $h$-fix (right) RTFIM on the square lattice. For both models $z'$ diverges at the critical point, whose position is affected by strong finite-size and temperature effects (see Sec. 5.3). However, the convergence behaviour seems different from the data.

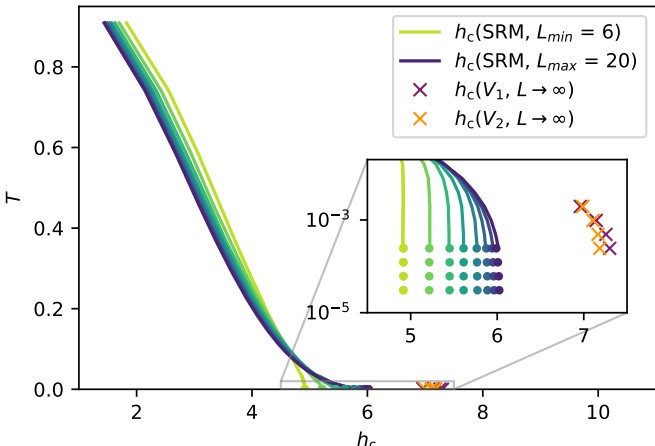

Figure 17: The mean of the distribution of pseudo-critical points extracted using the sample-replication method (SRM) is shown for a large range in temperature. For comparison, the critical point extracted using the intersections of Binder ratios $V_1$ and $V_2$ is displayed. The inset shows a cutout for very small temperatures on a log scale.

## 5.3 Temperature effects

During our simulations we observed that the influence of temperature is much more important than we anticipated. In pure systems, when the relation between the typical energy and length scale is algebraic, the convergence to zero temperature of a system of length $L$ can be estimated by $\beta_{\min} \sim 1/\Delta \sim L^z$. In many cases (e. g. $\mathcal{O}(N)$ symmetry breaking quantum phase transitions [2]), $z$ is one and therefore the relation is linear. However, activated scaling introduces exponential behaviour to the convergence in temperature. To better understand the convergence, we computed the distribution of critical points in a wide range of temperatures using the sample-replication method. The mean critical point for the same set of system sizes

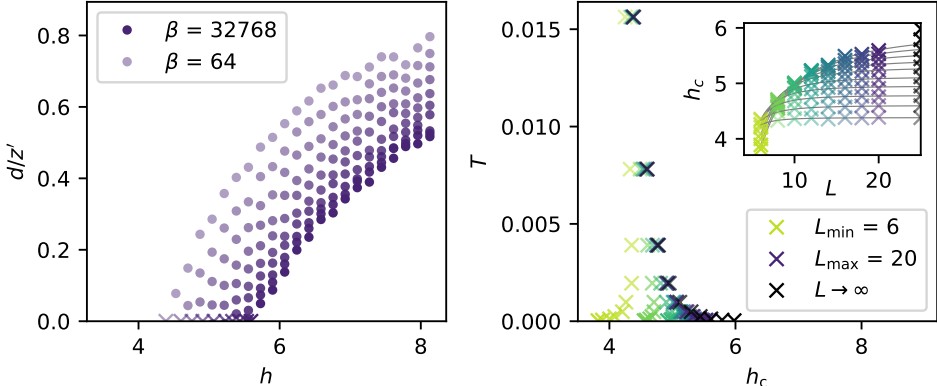

Figure 18: Left: The temperature dependence of the exponent $d/z'$ for the $L = 20$ $h$-box RTFIM on the square lattice is shown. The pseudo-critical point defined by $d/z' = 0$ strongly depends on $T = 1/\beta$. Right: The pseudo-critical points defined by $d/z' = 0$ for various $L$ and $\beta$ are extrapolated to $L = \infty$ and $T = 0$. Data points with higher temperature are displayed with decreasing saturation. The pseudo-critical points are extrapolated to $L = \infty$ in the inset. In the main plot the temperature-dependent pseudo-critical points are displayed.

used before is shown in Fig. 17. For large temperatures the curves look as if they would cross with the $x$-axis at approximately $h_c = 4.2$ being consistent with the result of Ref. [38, 39]. However, it stands out that close to $T = 0$, we see exponential behaviour, that looks fundamentally different than the behaviour for large $T$. In the inset you can see a cutout of our data at very small temperatures on a logarithmic scale. Even though we cooled the systems down to $1/T = \beta = 2^{15}$, one may argue that the largest system sizes are not perfectly converged yet. This very unusual temperature behaviour shows how important it is to cool down the system in a controlled manner.

The distribution of local susceptibilities is even more affected by temperature effects. In Fig. 18 (left) the exponent $d/z'$ is shown for the $L = 20$ $h$-box model for different temperatures. Although other observables are converged in temperature at this point, $d/z'$ is obviously not. It follows that this observable is not well suited to determine the critical point at comparable temperatures. This may explain the discrepancy between the critical points extracted in Refs. [38,39] and the critical point determined in this work or in the study of Ref. [40]. In Fig. 18 (right) we attempt to extrapolate the points $d/z'(L, \beta) = 0$ to the limit $L = \infty$ and $\beta = \infty$. First, we extrapolate to $L \to \infty$ using an algebraic fit (inset of Fig. 18 (right)). Then we plot the resulting temperature-dependent critical point over the temperature (main plot of Fig. 18 (right)). We see that the dependence of the critical point on temperature extracted with this method is non-trivial and hard to extrapolate. However, it seems that the critical point in the limit of large $L$ and small $T$ tends to shift towards larger values, which is in line with our previous results.

# 6 Conclusion

We investigate the critical behaviour of the RTFIM with different types of disorder in one and two dimensions using the SSE QMC approach. In order to determine zero-temperature properties, we use a rigorous scheme for cooling down and check the convergence in temperature for all observables separately. Due to exponentially small energy gaps, temperature must be kept at very low values up to $1/T = \beta = 2^{17}$. The non-self-averaging nature of the

IDFP [27, 28, 41, 49, 80] poses a great challenge for any numerical simulation of finite systems. The data presented in this work demanded extensive simulations in the order of $10^6$ CPU-hours.

From the generated data critical points and exponents were extracted by various finite-size scaling approaches. We exploited the behaviour of Binder ratios to determine an accurate estimate of the critical points. Furthermore, a sample-replication method was employed to obtain the distribution of pseudo-critical points of finite systems. By analysing the scaling properties of the distribution, the exponents $\nu_s$ and $\nu_w$ as well as the critical point $h_c$ were derived. The averaged magnetisation is used to determine the exponents $\beta/\nu_{av}$ and $\nu_{av}$.

We also discussed the strong temperature dependence of observables like the local susceptibility, which is used to determine the dynamic exponent $z'$. Besides the strong temperature dependence caused by activated scaling, we found corrections to finite-size scaling depending on the type of disorder. We conclude that it is beneficial to choose the type of disorder as strong as possible to converge to the exponents of the IDFP already on smaller finite systems. In one dimension, we gauged the quality of our methods and results with previous analytical [28] and numerical findings [52] of quantum critical properties. For the two-dimensional RTFIM on the square lattice with $h$-box disorder, the obtained critical exponents are in line with results from other numerical studies [32, 35, 36, 40, 54, 78, 79]. For the same model with $h$-fix disorder we observe strongly system-size dependent critical exponents, which is in line with our findings in one dimension. The exponents tend to converge towards the ones of the 2D-IDFP determined by SDRG studies [32, 35, 36, 54, 78]. However, in our analysis we cannot rule out, whether they might converge towards a different set of critical exponents being part of a different universality class as indicated by Ref. [40]. We provide an unbiased estimate for the location of the critical point and resolve the inconsistency of varying values in literature, which originates from an insufficient temperature convergence of observables.

We stress that our analysis is able to capture quantum critical properties well. It is not restricted in terms of dimension and lattice geometry as long as no sign problem exists in the SSE QMC. For the RTFIM, this even includes geometrically frustrated interactions. It would therefore be highly interesting to investigate the effects of quenched disorder on order-by-disorder [4–6, 81] and disorder-by-disorder [4, 5, 7, 82] mechanisms, which are intriguing physical phenomena arising in frustrated systems due to extensive ground-state spaces yielding novel states of quantum matter. In a similar spirit, also the interplay of long-range interactions and quenched disorder in low-dimensional, potentially frustrated quantum magnets is a promising future research direction. In both cases, our understanding is still in its infancy and novel physical phenomena are expected to arise.

# Acknowledgments

The authors thank Federico Becca, Ferenc Igloi, Istvan Kovacs and Cecile Monthus for fruitful e-mail exchange. We thankfully acknowledge the scientific support and HPC resources provided by the Erlangen National High Performance Computing Center (NHR@FAU) of the Friedrich-Alexander-Universität Erlangen-Nürnberg (FAU).

**Funding information** This work was funded by the Deutsche Forschungsgemeinschaft (DFG, German Research Foundation) - Project-ID 429529648 - TRR 306 QuCoLiMa (Quantum Co-operativity of Light and Matter). We acknowledge the support by the Munich Quantum Valley, which is supported by the Bavarian state government with funds from the Hightech Agenda Bayern Plus. The hardware of NHR@FAU is funded by the German Research Foundation DFG.

**Data availability** The raw data used in this work as well as processed data are provided in Ref. [77].

# A  Verification

## A.1  Sample-replication method: Comparison with analytic results

Numerical methods should always be compared with analytical methods in order to identify any implementation errors. The implementation of the SSE quantum Monte Carlo method itself was verified with exact diagonalisation. Besides that, we would also like to check the method we use to determine sample-dependent pseudo-critical points. To verify the sample-replication method, we have the opportunity to compare with the analytic result from Ref. [48] for the one-dimensional RTFIM, i. e. the relation stated in Eq. (4) leading to Eq. (37), which is only valid at the critical point. Note that in the derivation of Eq. (4) a boundary term is neglected, which vanishes in the thermodynamic limit. For finite systems the prediction using Eq. (37) is therefore expected to slightly deviate from the true pseudo-critical point. In Fig. 19 we use 20 sets of bond- and field-strengths $\{J_{i,i+1}, h_i\}$ and determine the pseudo-critical point both using the sample-replication method with data generated by SSE QMC and by inserting into Eq. (37). It can be seen that the pseudo-critical points of the different methods coincide quite well. The deviations between the data points are easily explained. Statistical fluctuations appear since we do not focus on high accuracy single curves in our simulation, i.e. large numbers of Monte Carlo steps, but rather aim to simulate as many disorder realisations as possible. In the end, averaging over disorder realisations will also average over statistical inaccuracies [58]. Besides this, there is also a systematic shift visible in the data, which becomes smaller as the system size increases (comparing $L = 6$ (left) with $L = 20$ (right) in Fig. 19). We expect this to be the contribution of the neglected boundary term in the derivation of Eq. (4), which leads to the shifting of $\left[\log(\tilde{h}_c(L))\right]$ elaborated on in the main body of the paper. This is consistent with the assumption that the contribution from the boundary term vanishes for very large system sizes. However, it cannot be ruled out that our numerical protocol to determine the pseudo-critical point from $\Phi(h, L)$ leads to an additional error that disappears in the limit of large systems, as $\Phi(h, L)$ is less rounded with increasing $L$.

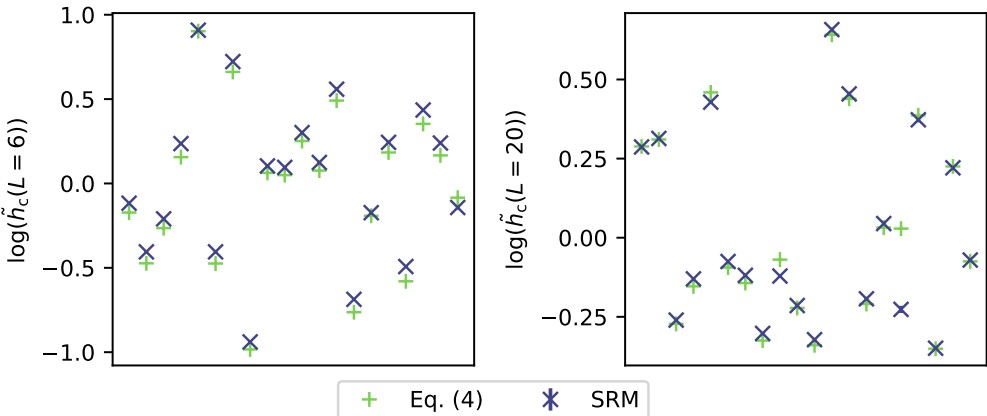

Figure 19: Sample-dependent pseudo-critical points extracted using the sample-replication method (SRM) for the $h$-box RTFIM on the linear chain. We can compute the same critical points using Eq. (37) and the bond- and field- strengths $\{J_{i,i+1}, h_i\}$ used for the simulation.

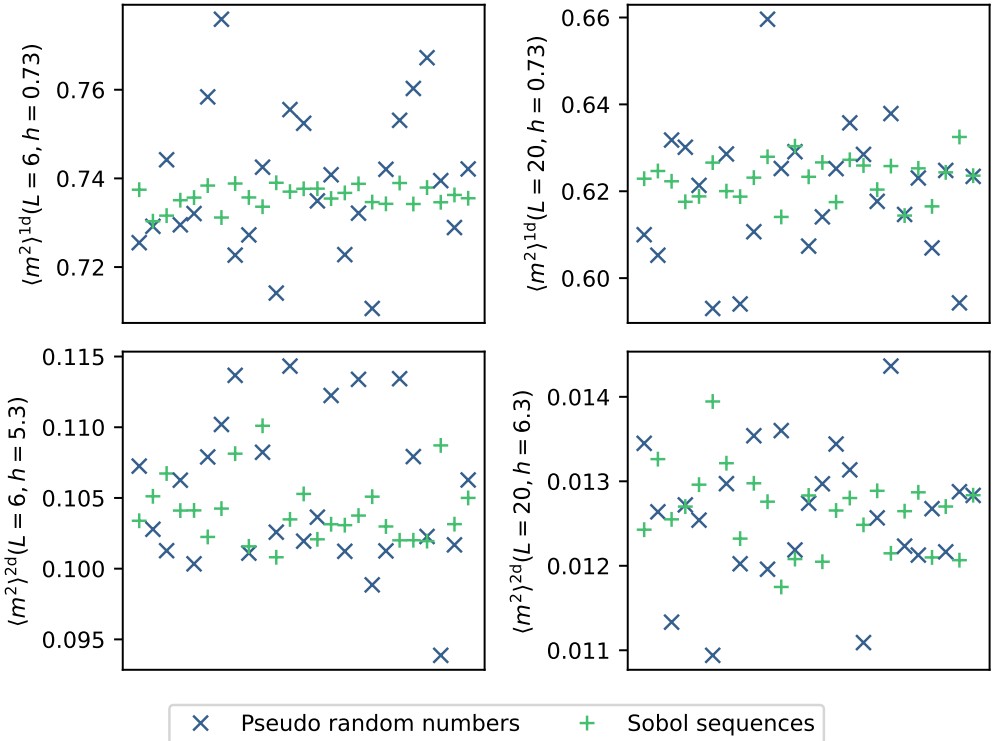

Figure 20: Magnetisation at a single value of $h$ for the $h$-box RTFI chain (upper plots) and the $h$-box RTFIM on a square lattice (lower plots) using pseudo-random numbers and Sobol sequences for the bond- and field- strengths $\{J_{i,j}, h_i\}$. Every data point is averaged over 128 disorder configurations. The magnetisation simulated using Sobol sequences converges faster in the number of samples than simulated with pseudo-random numbers. With increasing dimension of the disorder configuration space (here: $s = 12, 40, 108, 1200$) the advantage of the Sobol sequences seems to decrease, but is still present.

### A.2 Advantage of Sobol sequences

The discrepancy of a low-discrepancy sequence of $N$ samples is limited by [61, 83]

$$D_N < c_s \frac{\log(N)^s}{N}, \tag{A.1}$$

where $s$ is the dimension of the configuration space and $c_s$ is a constant dependent of the actual sequence. One expects that low-discrepancy sequences perform particularly well for small $s$ with a convergence of approximately $\sim 1/N$ compared to $\sim 1/\sqrt{N}$ for pseudo-random numbers. Sobol sequences are low-discrepancy sequences that also perform well in higher dimensional spaces [61]. While the advantage of quasi-random numbers compared to pseudo random numbers for the integration of functions in $s$-dimensional spaces is generally known [61], it is difficult to estimate the advantage it has on the observables we are interested in. In Fig. 20 we present the magnetisation of the $h$-box RTFI chain and the $h$-box RTFIM on the square lattice for a fixed $h$ each. Each data point is averaged over 128 disorder configurations drawn from Sobol sequences or pseudo random numbers. Besides that the simulation time and number of Monte Carlo steps is the same for both sets. We see that the magnetisation of Sobol sequences converges faster than for pseudo random numbers for all systems. However, the advantage decreases with the dimension of the configuration space. Note that only very

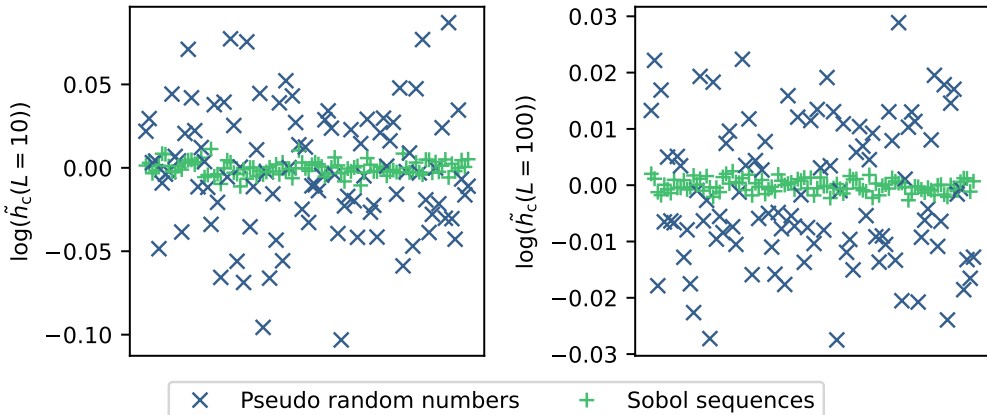

Figure 21: Critical point determined from Eq. (37) using a pseudo-random number generator and Sobol sequences for the bond- and field- strengths $\{J_{i,i+1}, h_i\}$. Every data point is averaged over 128 disorder configurations. We see that observables simulated by using Sobol sequences converge much faster in the number of samples than simulated with pseudo-random numbers.

few configurations were averaged here. With increasing $N$ the difference should become even clearer. In Fig. 21 we show the pseudo-critical points $\left[\log(\tilde{h}_c(L))\right]$ for the $h$-box RTFI chain calculated using Eq. (37). The same comparison could have been made with data generated by SSE and the sample-replication method, however, in order to exclude influence by further statistical errors and not to waste simulation time, we limit ourselves to Eq. (37). Every data point is averaged over 128 disorder realisations. We observe that in the case when the bond- and field-strength $\{J_{i,i+1}, h_i\}$ are drawn from Sobol sequences, the critical points converge much faster than when drawn from a pseudo-random number generator. It is striking that the advantage does not visibly decrease, when we increase the system size, i.e. the dimension of the disorder configuration space (see Fig. 21 (right)).

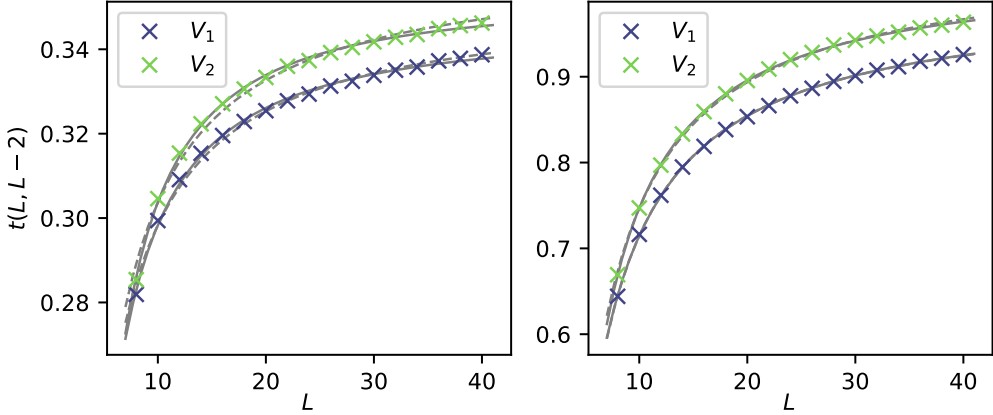

Figure 22: Intersections of neighboring Binder ratios ($L$ and $L-2$) are determined for $h$-fix (left) and $h$-box (right) disorder of the one-dimensional RTFIM. Here larger system sizes are evaluated than in the main body of the paper. Therefore we use the method described in App. B to calculate Binder ratios from magnetisation curves very efficient. The solid and dashed lines are algebraic and $1/L$ fits to the data points respectively.

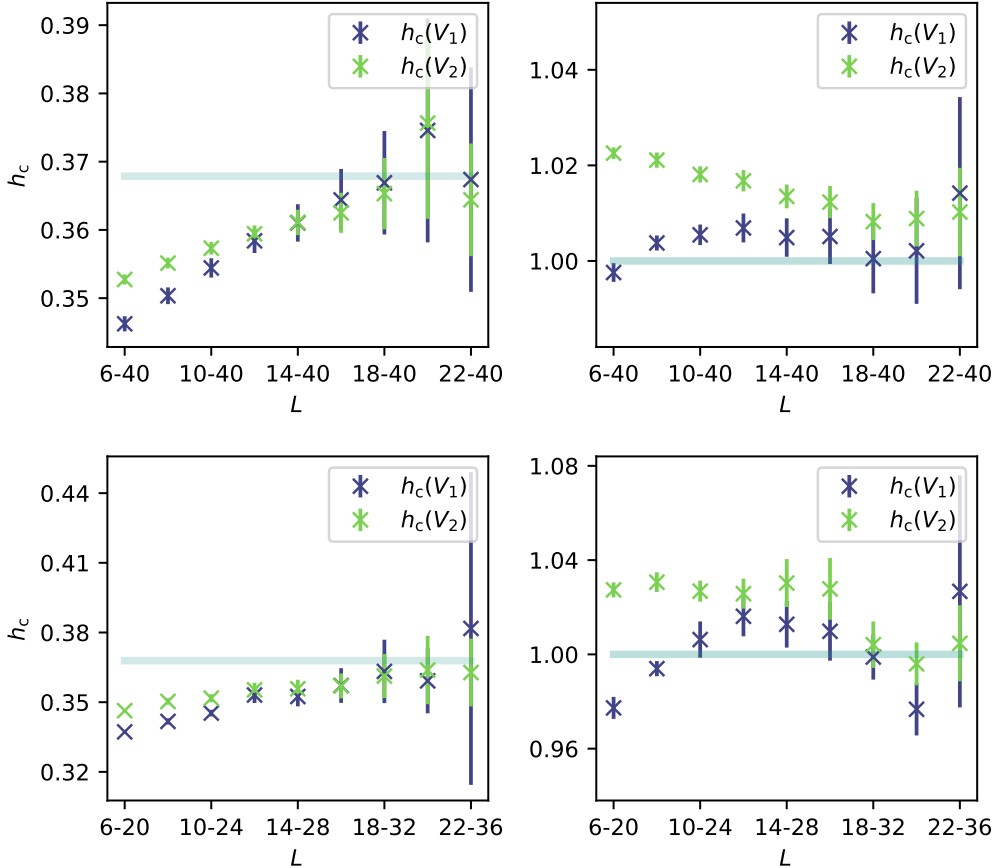

Figure 23: Using the data points shown in Fig. 22, we analyse the $L$-dependency of the critical point $h_c$ extracted by this method. The solid line indicated the expected critical point for the respective system [48], i. e. the $h$-fix chain (left) and $h$-box chain (right). In the upper plots smaller system sizes are successively removed from the fits, in the lower plots a set of 8 system sizes is considered for every fit.

## A.3 Results for the one-dimensional model on larger system sizes

To verify the results for the one-dimensional model with small system sizes $L = 6-20$ presented in the main body of the paper, we have additionally simulated larger systems $L = 6-40$. To exclude any influence of temperature, the data for the larger system sizes is not determined by the SSE QMC method, but by the Jordan-Wigner method, which is introduced in App. B. Every data point is averaged over at least 200000 disorder realisations. In Fig. 22 one can see the intersections $t(L, L-2)$ of the Binder ratios $V_1$ and $V_2$ for the $h$-fix and $h$-box RTFI chain. The extrapolation leads to $h_c(V_1) = 0.346$ and $h_c(V_2) = 0.353$ for the $h$-fix model and $h_c(V_1) = 0.998$ and $h_c(V_2) = 1.023$ for the $h$-box model. These values are consistent with the results we obtained using smaller system sizes and are even closer to the exact values $h_c = 1/e$ and $h_c = 1$. We further analyse the $L$-dependency of the extracted critical points in Fig. 23 in two different ways by successively excluding smaller systems. We can see that the deviation to the exact critical point decreases with increasing system size in the case of the $h$-fix model. In the case of the $h$-box model the convergence is less clearly seen. However, the critical points nevertheless seem to be consistent with $h_c = 1$. It is also noticeable that the error on the fit increases with increasing system size, although the number of disorder realisations is the same for all system sizes. This is due to the fact that the binder ratios intersect more shallowly with increasing system size, which leads to a greater statistical error.

# B  Jordan-Wigner calculation of correlation functions for the 1D-RTFIM

As first described in Ref. [46], one can solve the one-dimensional TFIM by applying a Jordan-Wigner transformation [84]. The resulting quadratic fermionic Hamiltonian can be diagonalised in momentum space using a Bogoliubov transformation. When disorder is present, the problem for a TFIM on a chain with $N$ sites only reduces to a Bogoliubov transformation in real space. This was first utilised in Ref. [51] to numerically study the RTFIM. We repeat the most important steps.

## B.1  Diagonalising the Hamiltonian

First, the spin operators are mapped to hardcore-boson operators with the Matsubara-Matsuda transformation [85]

$$\sigma_z^i = a_i^\dagger + a_i \,, \tag{B.1}$$

$$\sigma_x^i = 1 - 2a_i^\dagger a_i \,. \tag{B.2}$$

Next one applies the Jordan-Wigner transformation to fermionic operators $c_i$

$$a_i^\dagger = c_i^\dagger \exp\left[-i\pi \sum_{j=1}^{i-1} c_j^\dagger c_j\right] \,. \tag{B.3}$$

In the representation of fermionic operators, the TFIM takes the form

$$\mathcal{H} = -\sum_{i=1}^N h_i\left(1 - 2c_i^\dagger c_i\right) - \sum_{i=1}^{N-1} J_i(c_i^\dagger - c_i)(c_{i+1}^\dagger + c_{i+1}) + J_N(c_N^\dagger - c_N)(c_1^\dagger + c_1)\exp[i\pi\mathcal{N}]\,, \tag{B.4}$$

where $\mathcal{N} = \sum_{i=1}^N c_i^\dagger c_i$ is the number of fermions. Hence, in the sector of even parity $\exp[i\pi\mathcal{N}] = 1$. Introducing

$$\Psi^\dagger = (c_1^\dagger, c_2^\dagger, \dots, c_N^\dagger, c_1, c_2, \dots, c_N)\,, \tag{B.5}$$

which satisfies fermionic anti-commutation relations

$$\{\Psi_i^\dagger, \Psi_j\} = \delta_{ij}\,, \quad \text{and} \quad \{\Psi_i^\dagger, \Psi_j^\dagger\} = 0\,, \tag{B.6}$$

one can write the Hamiltonian as

$$\mathcal{H} = \Psi^\dagger \tilde{H} \Psi\,, \tag{B.7}$$

where $\tilde{H}$ is a $2 \times 2$ block matrix

$$\tilde{H} = \begin{pmatrix} A & -B \\ B & -A \end{pmatrix}\,. \tag{B.8}$$

The matrix elements are

$$\begin{aligned} A_{i,i} &= h_i\,, \\ A_{i,i+1} &= A_{i+1,i} = J_i/2\,, \\ B_{i,i+1} &= -B_{i+1,i} = J_i/2\,. \end{aligned} \tag{B.9}$$

The energies of $\mathcal{H}$ can now be easily obtained by diagonalising

$$\tilde{H} = V^\dagger D V\,. \tag{B.10}$$

We assume ordering of eigenvalues in ascending order. The eigenvalues in $D$ come in pairs of equal magnitude but different sign. The corresponding eigenvectors in $V$ are complex conjugates of another. We define $\epsilon_\mu$ as the $N$ positive energies in $D$ and operators $\gamma_\mu^\dagger$ equal to the eigenvectors in $V$ with positive energies. Then one can write the Hamiltonian in the normal form

$$\mathcal{H} = \sum_{\mu=1}^{N} \epsilon_\mu (\gamma_\mu^\dagger \gamma_\mu - \gamma_\mu \gamma_\mu^\dagger), \tag{B.11}$$

and the ground-state energy is

$$E_0 = -\sum_{\mu=1}^{N} \epsilon_\mu, \tag{B.12}$$

in the even-parity sector.

## B.2 Spin-spin correlations

The calculation of spin-spin correlation functions demands more effort. Inserting the Jordan-Wigner transformation, we get

$$C_{i,j} \equiv \left\langle \sigma_i^z \sigma_j^z \right\rangle = \left\langle (c_i^\dagger + c_i) \exp\left[-\mathrm{i}\pi \sum_{l=1}^{j-1} c_l^\dagger c_l\right] (c_j^\dagger + c_j) \right\rangle, \tag{B.13}$$

where $\langle \ldots \rangle$ denotes ground-state averaging. Apart from Ref. [51], a good description for the evaluation of this expectation value is given in Ref. [86]. Defining

$$\begin{aligned} A_l &= c_l^\dagger + c_l, \\ B_l &= c_l^\dagger - c_l, \end{aligned} \tag{B.14}$$

one finds

$$C_{i,j} = \left\langle B_i \left(A_{i+1}B_{i+1} \cdots A_{j-1}B_{j-1}\right) A_j \right\rangle. \tag{B.15}$$

Using Wick's theorem, this can be decomposed into a sum of products of two-point correlations [51, 86]. One needs

$$\begin{aligned} \left\langle A_i A_j \right\rangle &= \delta_{ij}, \\ \left\langle B_i B_j \right\rangle &= -\delta_{ij}, \\ \left\langle B_i A_j \right\rangle &= -\left\langle A_j B_i \right\rangle =: G_{i,j}, \end{aligned} \tag{B.16}$$

to evaluate

$$C_{i,j} = \det X^{i,j}, \tag{B.17}$$

with

$$X_{s,r}^{i,j} = G_{i-1+s,j+r}, \tag{B.18}$$

and $s = r = 1, \ldots, j-i$. To express $G_{i,j}$ by the eigenvectors $V$, we need the $N \times N$ blocks

$$V = \begin{pmatrix} V_{11} & V_{12} \\ V_{21} & V_{22} \end{pmatrix}. \tag{B.19}$$

Then

$$G_{i,j} = 1 - 2 V_{12} (V_{12} + V_{21})^\dagger = -(V_{12} - V_{21})(V_{12} - V_{21})^\dagger, \tag{B.20}$$

is a product of unitaries and also unitary. This way one can calculate $m^2$.

### B.3 Four-point correlations

To calculate Binder ratios with the Jordan-Wigner approach, one needs the four-point correlation functions

$$C_{i,j,k,l} \equiv \left\langle \sigma_i^z \sigma_j^z \sigma_k^z \sigma_l^z \right\rangle . \tag{B.21}$$

Those were not calculated in Ref. [51] and, to the best of our knowledge, this is done for the first time here for the RTFIM. In fermion operators $C_{i,j,k,l}$ takes the form

$$C_{i,j,k,l} = \left\langle (c_i^\dagger + c_i) \exp\left[ -i\pi \sum_{r=1}^{j-1} c_r^\dagger c_r \right] (c_j^\dagger + c_j)(c_k^\dagger + c_k) \exp\left[ -i\pi \sum_{s=k}^{l-1} c_s^\dagger c_s \right] (c_l^\dagger + c_l) \right\rangle$$

$$= \left\langle B_i \left( A_{i+1} B_{i+1} \cdots A_{j-1} B_{j-1} \right) A_j B_k \left( A_{k+1} B_{k+1} \cdots A_{l-1} B_{l-1} \right) A_l \right\rangle . \tag{B.22}$$

When one compares this with Eq. (B.15), it becomes apparent that this expression can again be expressed as a determinant of a matrix, i.e.

$$C_{i,j,k,l} = \det Y = \begin{pmatrix} Y_{11} & Y_{12} \\ Y_{21} & Y_{22} \end{pmatrix}, \tag{B.23}$$

with

$$\begin{aligned} Y_{11,s,r}^{i,j,k,l} &= G_{i-1+s,j+r}, \quad \text{for} \quad s = r = 1, \ldots, j-i, \\ Y_{22,s,r}^{i,j,k,l} &= G_{k-1+s,k+r}, \quad \text{for} \quad s = r = 1, \ldots, l-k, \end{aligned} \tag{B.24}$$

and

$$\begin{aligned} Y_{12,s,r}^{i,j,k,l} &= G_{i-1+s,k+r}, \quad \text{for} \quad s = 1, \ldots, j-i, \quad r = 1, \ldots, l-k, \\ Y_{22,s,r}^{i,j,k,l} &= G_{k-1+s,i+r}, \quad \text{for} \quad s = 1, \ldots, l-k, \quad r = 1, \ldots, j-i. \end{aligned} \tag{B.25}$$

All that is left is the evaluation of determinants to obtain $m^4$.

### B.4 Evaluation of determinants

Diagonalisation of $\tilde{H}$ and calculation of $G_{i,j}$ only has complexity of $\mathcal{O}(N^3)$. Evaluation of one determinant for $C_{i,j}$ itself has complexity $\mathcal{O}(N^3)$. A naive calculation of $m^2$ would thus have $\mathcal{O}(N^5)$ complexity. If one performs a LU decomposition of $X^{i,j}$ without pivoting, one does not only obtain $C_{i,j}$ but also $C_{i,s}$ for $i < s < j$. This way, the complexity reduces to only $\mathcal{O}(N^4)$. Here, one has to note that it is not a priori clear if the LU decomposition without pivoting is stable. Stability is given if all leading principal minors are non-zero or, from a numerical perspective, not too small. Since we are close to criticality, correlations fall off slowly and this condition is fulfilled for almost all disorder configurations. A naive calculation of $m^4$ has complexity $\mathcal{O}(N^7)$. Using LU decompositions we end up with complexity $\mathcal{O}(N^6)$. The theoretical complexity can be improved further using rank-1 updates for the LU-factorisation. For $m^2$ one only needs one full LU decomposition then and $N-2$ rank-1 updates of complexity $\mathcal{O}(N^2)$. Similarly, for $m^4$ the performance can be improved by one factor of $N$. However, for the system sizes considered in our paper, this performance gain did not show up. The reason is that the runtime of LU decomposition was strongly dominated by the $N^2$-term. Because of that we used the LU routines for our calculations. If one is interested in very large systems, usage of rank-1 updates will become beneficial.

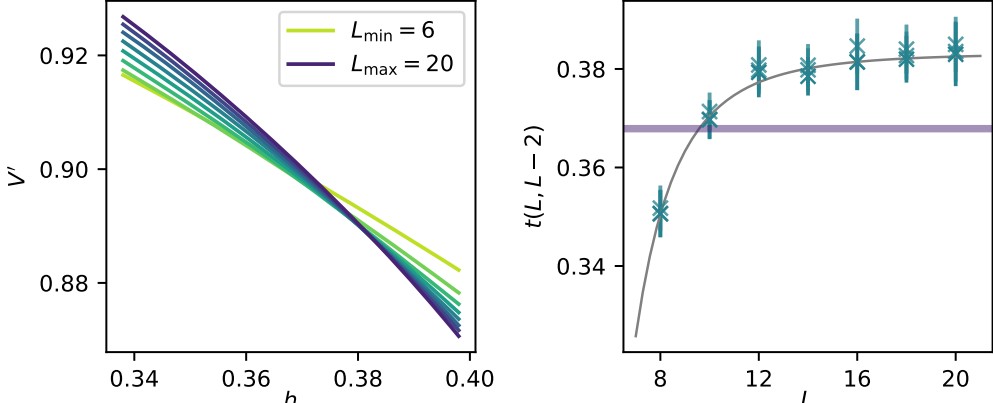

Figure 24: Left: $V'$ for different system sizes for the $h$-fix RTFI chain. Right: Intersections of $V'$ scaled to $L \to \infty$. The critical point in the thermodynamic limit $h_c = 1/e$ is indicated by a straight line. The faded data points are the intersection points from simulation with higher temperature ($\times 2$, $\times 4$ and $\times 8$ with decreasing saturation) measured during the beta doubling procedure.

## C  Further finite-size scaling methods for disordered systems

### C.1  Binder ratios

Besides the definitions of $V_1$ and $V_2$, which we considered in the main body of the article, there are more ways to define Binder-like ratios, that should intersect at the critical point. In Ref. [72], $V'$ was defined as

$$V' = \frac{1}{2}\left( 3 - \frac{\left[ \langle m^4 \rangle \right]}{\left[ \langle m^2 \rangle^2 \right]} \right), \tag{C.1}$$

choosing a different approach of averaging. Note that for pure systems there would be no difference between $V'$ and $V_1$ or $V_2$. $V'$ is depicted in Fig. 24 (left) for the $h$-fix RTFI chain, the extracted intersection points $t(L, L-2)$ are shown in the right part of the figure. We use the same raw data for $V'$ as we did for $V_1$ and $V_2$. Scaling the intersections to $L \to \infty$ massively overestimates the critical point. Even worse, we see that most of the intersections lie above the critical point and scale towards higher values. Assuming monotonic scaling, these intersection points are not compatible with the analytically known critical point $h_c = 1/e$. Another way to define a Binder-like quantity is given in Ref. [73] by

$$U_1 = \frac{\left[ \langle |m| \rangle^2 \right]}{\left[ \langle |m| \rangle \right]^2}. \tag{C.2}$$

Note that $U_1$ is always equal to 1 in a pure system and can therefore only be defined for disordered systems. Analogous to $V'$, we present $U_1$ in Fig. 25 (left) using the same data set. On the right, the respective intersection points are depicted together with the expected critical point $h_c = 1/e$. Scaling to $L \to \infty$ does again not provide a compatible critical point, however, this time we cannot rule out that scaling for larger systems changes in a monotonous way towards the right critical point.

To conclude, we again want to stress the importance of averaging for disordered systems. Even though these definitions of Binder ratios seem to be similar, they show completely different

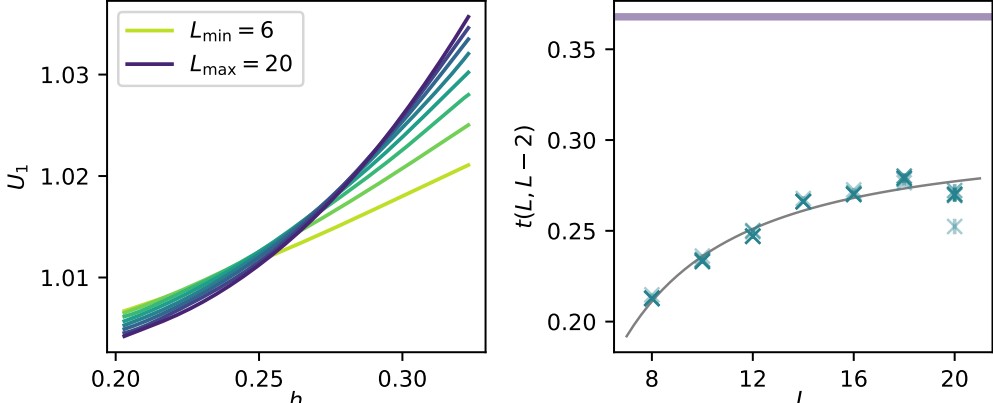

Figure 25: Left: $U_1$ for different system sizes for the $h$-fix RTFI chain. Right: Intersections of $U_1$ scaled to $L \to \infty$. The critical point in the thermodynamic limit $h_c = 1/e$ is indicated by a straight line. The faded data points are the intersection points from simulation with higher temperature (×2, ×4 and ×8 with decreasing saturation) measured during the beta doubling procedure.

scaling behaviour and may also lead to different estimates for the critical point in the thermodynamic limit. Furthermore, the right choice seems to be model dependent as the definitions presented above lead to improved results for the site-diluted Heisenberg models in Ref. [72], but failed in the case of the RTFI chain. It is an open question which type of averaging is beneficial for which model. Therefore, the best approach is to compare with analytical results when possible. We also looked at $V'$ and $U_1$ for the two-dimensional RTFIM (not shown here) and received estimates for the critical point in the same order of magnitude compared with the methods presented in the main part of the article. However, since the verification in one dimension failed, we did not include these results here.

### C.1.1 Imaginary-time integrated Binder ratio

So far, we used in our evaluation the $\tau = 0$ magnetisation to calculate observables. As stated in Sec. 3.1, we have also access to imaginary time integrated observables with the SSE QMC method. Using Eq. (19), we can define an imaginary-time integrated Binder ratio as follows

$$V_{\text{int}} = 1 - \left[ \frac{\langle m_{\text{int}}^4 \rangle}{3 \langle m_{\text{int}}^2 \rangle^2} \right]. \tag{C.3}$$

We choose the constant differently here to better compare with the results of Ref. [40] scaling only the amplitude of the data points. In contrast to the Binder ratios considered so far, we do not expect this quantity to intersect at the critical point, but all curves plotted over inverse temperature $\beta$ should have the same (universal) amplitude at the critical point [39, 40]. In Fig. 26 $V_{\text{int}}$ is presented for several values of $h$. The condition that the maxima of all curves have the same value is fulfilled best at $h = 8.892$ (lower right plot), which is far away from the critical point we extract using the methods in the main body of the article. As for any other observable we discussed, we also assume for $V_{\text{int}}$ that there are strong corrections to scaling. In Ref. [40] the maximum of their two largest system sizes $L = 32$ and $L = 36$ was compared to determine the critical point to be $h_c = 7.52$, since smaller systems are affected stronger by finite-size corrections. Comparing our largest system sizes $L = 18$ and $L = 20$ would lead to a critical point of approximately $h_c = 7.780$ (see lower left plot). For comparison we also displayed the curves for $h = 7.224$, which is close to the critical point we claim in this work,

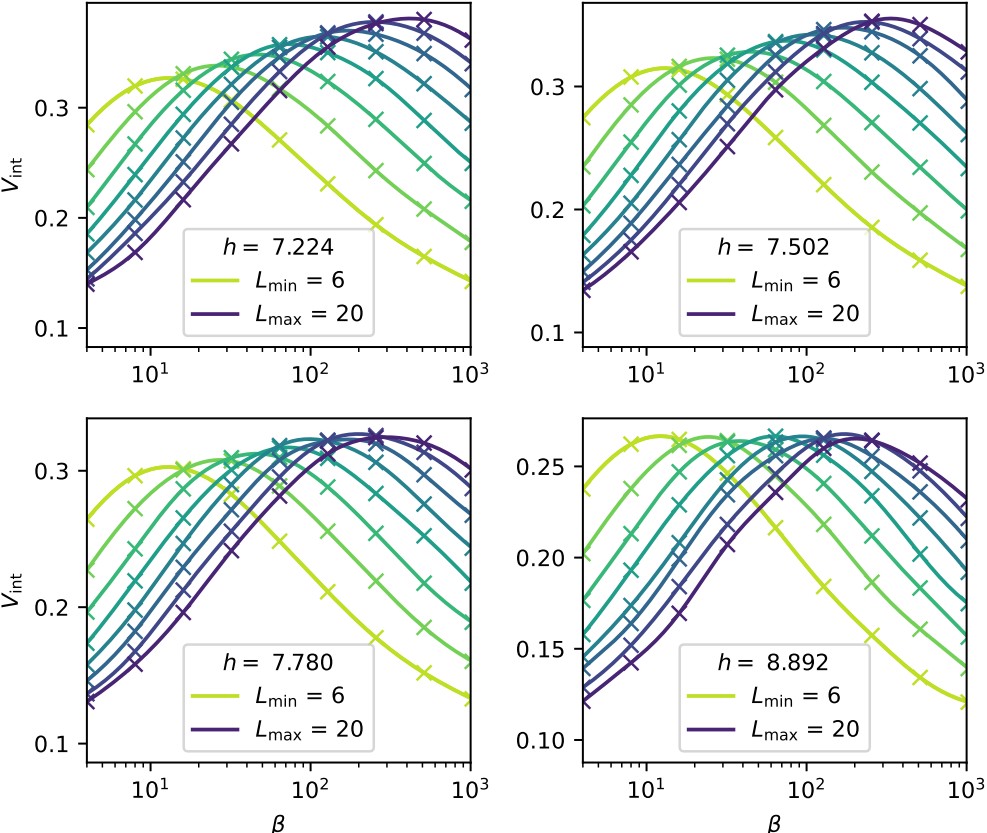

Figure 26: The imaginary time integrated Binder ratio $V_{\text{int}}$ is shown for different values of $h$ for the $h$-box RTFIM on the square lattice. One would expect the maximum of the curves to be independent of the system size at the critical point.

and $h = 7.502$, which is close to the critical point claimed in Ref. [40] in the upper two plots of Fig. 26. Both seem to not fulfil the condition of having the same maximum value. We therefore conclude that probably our system sizes are too small to determine an accurate estimate for the critical point using this method. However, it is questionable if increasing the linear system sizes by a factor of two would solve this problem.

## C.2 Data collapse

As already described in the main section, it turned out that a data collapse is not the optimal method for determining the critical point and the critical exponents for the RTFIM on small systems. Nevertheless, we would like to briefly introduce the method and present the results we get from it for our data. For this purpose, we use the data for the $h$-box RTFI chain up to $L = 40$, which were determined using the Jordan-Wigner method (see App. B). Following Eq. (23), in this method the function

$$m^2(r, L) = L^{-2\beta/\nu} f_{m^2}(r L^{1/\nu}), \tag{C.4}$$

is fitted to the magnetisation curves $\left[\langle m^2(r, L)\rangle\right]$ for all $L$ simultaneously. The scaling function $f_{m^2}$ is assumed to be a polynomial of degree 4. Whether the data collapse fits the data well can be shown by rescaling $\left[\langle m^2(r, L)\rangle\right]$ with a factor of $L^{2\beta/\nu}$ in $y$-direction and a factor of $L^{1/\nu}$ in $x$-direction. Since we analyse averaged magnetisation curves, we assume $\nu = \nu_{\text{av}}$. Applying this procedure to our data in the range of the small system sizes investigated in the

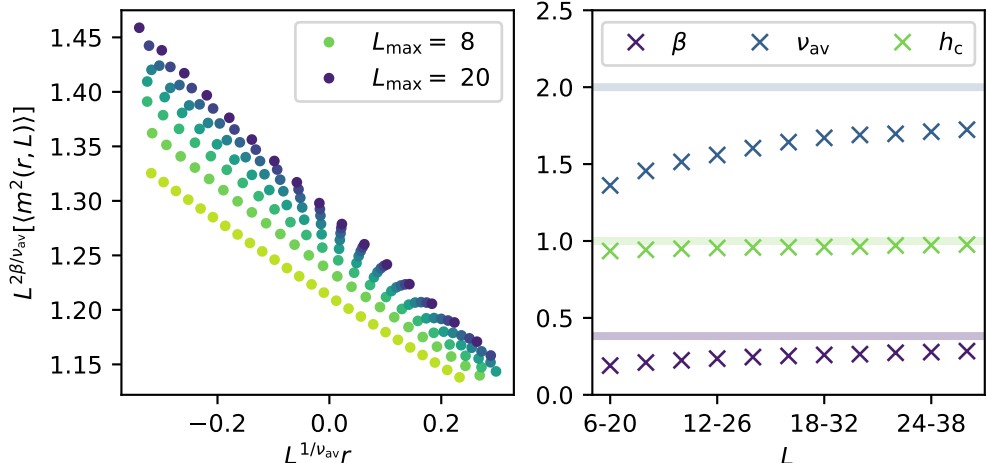

Figure 27: Left: Data collapse for the $h$-box RTFI chain fixing the exponents $\nu_{\text{av}} = 2, \beta = (3 - \sqrt{5})/2$ and the critical point $h_{\text{c}} = 1$ using small system sizes $L = 6 - 20$. Right: Fitting results from the data collapse method for free parameters $\nu_{\text{av}}, \beta$ and $h_{\text{c}}$. To see the $L$-dependency of the exponents and critical point we perform the data collapse for different sets of system sizes $L$.

main part, i.e. $L = 6 - 20$, leads to critical exponents $\beta = 0.190$, $\nu_{\text{av}} = 1.36$ and the critical point $h_{\text{c}} = 0.935$. This is rather far away from the known values in one dimension. In Fig. 27 (left) the rescaled magnetisation curves for $L = 6 - 20$ are shown using the correct exponents $\nu_{\text{av}} = 2, \beta = (3 - \sqrt{5})/2$ and the correct critical point $h_{\text{c}} = 1$. One can see that the data points do not collapse to one curve, but deviate especially for the smaller system sizes. Using the larger system sizes up to $L = 40$ we can determine the $L$-dependency of the critical point and exponents (see Fig. 27 (right)). Therefore we successively exclude smaller system sizes and include larger system sizes and perform the data collapse for each set of system sizes. We observe that the critical point is captured quite well with the method, but the critical exponents only converge very slowly towards the correct values.

## C.3 Corrections to scaling

Since corrections are prominent in the systems investigated in this work, we want to identify the leading corrections to scaling and apply them to our data. For pure systems, Ref. [75] provides a framework to extract and fit the leading corrections to observables. The idea is based on the fact that the free energy density $f$ can be described by a generalised homogeneous function (GHF) close to the critical point [63–65] and thermodynamic observables which are derived from it are also GHFs [65]. This is the basis of the standard finite-size scaling theory for pure systems. Besides the relevant couplings, there are also irrelevant couplings present in $f$, that vanish in the limit of large $L$. The approach of Ref. [75] takes into account the leading irrelevant coupling, i.e. the irrelevant coupling with largest exponent $y_{\text{max}} := -\phi/\nu$ to obtain a modified version of Eq. (23):

$$\mathcal{O}(r, L) = L^{-\omega/\nu}(1 + cL^{-\theta})f_{\mathcal{O}}(rL^{1/\nu} - dL^{-\phi/\nu}), \tag{C.5}$$

where $\theta = \min(\phi/\nu, 1)$, and $c$ and $d$ are non-universal constants. Following this form, the intersections of Binder ratios $t_i(L, nL)$ are no longer independent of $L$, but scale as follows [75]

$$t(L, nL) = \frac{cq_0}{q_1}\left(\frac{1 - n^{-\theta}}{n^{1/\nu} - 1}\right)L^{-1/\nu - \theta} - d\left(\frac{1 - n^{-\phi/\nu}}{n^{1/\nu} - 1}\right)L^{-(1+\phi)/\nu}, \tag{C.6}$$

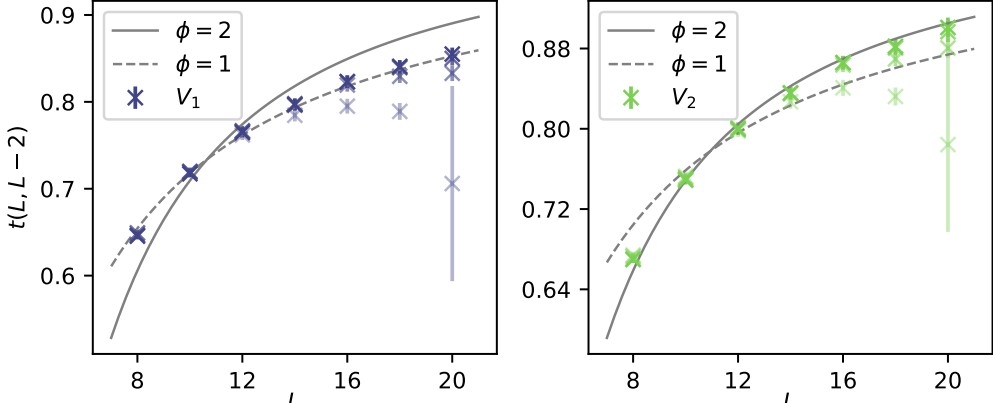

Figure 28: Intersections of binder ratios $V_1$ and $V_2$ for the $h$-box RTFI chain. Eq. (C.7) is fitted to the data point fixing $h_c = 1$ and $\nu = 2$ for two different values of the correction exponents $\phi$.

where $q_0$ and $q_1$ are further non-universal constants. For the intersection of neighbouring system sizes $t(L, L-2)$, we can rewrite Eq. (C.6) to

$$t(L, L-2) = \frac{cq_0}{q_1} \left( \frac{1 - \left( \frac{L-2}{L} \right)^{-\theta}}{\left( \frac{L-2}{L} \right)^{1/\nu} - 1} \right) L^{-1/\nu - \theta} - d \left( \frac{1 - \left( \frac{L-2}{L} \right)^{-\phi/\nu}}{\left( \frac{L-2}{L} \right)^{1/\nu} - 1} \right) L^{-(1+\phi)/\nu}. \tag{C.7}$$

It goes without saying that this form is way too complex to fit it to data without risking overfitting. Therefore, we applied Eq. (C.7) to the data for $V_1$ and $V_2$ for the $h$-box RTFI chain fixing all known constants, i.e. $h_c = 1$ and $\nu = \nu_{av} = 2$. Interestingly enough we receive $\phi \approx 1$ using $V_1$ and $\phi \approx 2$ using $V_2$. In Fig. 28 you can see fits with fixed $\phi = 1$ and $\phi = 2$ which only match the behaviour of one binder ratio each. It is striking that the choice of averaging influences the scaling power of the leading irrelevant coupling.

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
