# Peer review of "Quantum-critical properties of the one- and two-dimensional random transverse-field Ising model from large-scale quantum Monte Carlo simulations"

_SciPost Physics, doi:SciPost Phys. 17, 061 (2024)_

## Round 1 · Referee Report · Heiko Rieger (Referee 1) · 2024-4-19

Strengths

see report

Weaknesses

see report

Report

The authors consider the random transverse field Ising model in one and two space dimensions and study its critical properties using stochastic series expansion (SSE) quantum Monte Carlo (QMC) and finite size scaling. For 1d the known results for the critical point as well as the critical exponents are reproduced within statistical and extrapolation error bars, in addition new insights into convergence and finite size properties are revealed. In 2d the authors determine a new critical field value (for the box distribution), which is higher than previous numerical estimates and critical exponent values that agree within the error margins with previous values extracted from QMC simulations and SDRG calculations. Again, due to the enormous numerical effort made and the new methods applied in the data analysis, novel insights into convergence and finite size properties are revealed, paving the way for future numerical studies of disordered systems governed by an infinite randomness fixed point (IRFP).

The paper is very well written and discusses thoroughly all potential problems that occur in QMC studies of critical random systems with activated dynamics, meaning with exponentially diverging time scales at the critical point). This is an excellent work and in my view it presents a breakthrough in analysis of an extremely hard problem, namely the computational determination of the critical properties of higher-dimensional disordered systems that are supposedly governed by an infinite randomness fixed point, a previously-identified and long-standing research stumbling block in Sci. Post. expectation terms. It also fulfills *all* general acceptance criteria. Therefore, I recommend its publication in Sci. Post. in its present form after the following minor points have been considered:

1) To my knowledge, there is a zero-temperature version of SSE (T=0 SSE) which can be applied to the TFIM (Section 1.4.2 in Stochastic series expansion quantum Monte Carlo by Roger G. Melko). As discussed thoroughly by the authors, an extrapolation to T=0 (especially for the RTFIM) requires one to approach very low temperature, so it would be advantageous if such a T=0 SSE can capture T=0 properties without extrapolation. It would be useful if the authors could discuss briefly whether T=0 SSE captures the T=0-properties of the random TFIM or not, or whether the T=0 SSE is hard or impossible to be applied in the present case.

2) The SDRG for the two-dimensional RTFIM (ref. [35] in the paper) predicts that critical exponents of h-fix and h-box are equal, but ref. [40] concludes, also numerically with QMC, that they are different. Although the authors focus on the h-box distribution, exclusively, it would be useful if they found indications pointing in the same direction as in [40] or not.

3) p.4: “… rare regions with finite clusters with very special disorder configurations …”
I wonder why the authors denote these configurations as “very special”: the commn understanding is that these configurations comprise strongly coupled clusters, i.e. clusters (compact or fractal) that are locally in the ordered phase.

4) p.5: “The critical exponent of the dynamical scaling is given by psi=1/2”
Since usually the dynamic exponent is denoted as z I would more precisely write “… of the activated dynamic scaling …”

5) p.9: “causing different disorder configurations to converge in temperature at different β values”: I wonder whether similar convergence variations appear in the number of necessary MCS, N_MC, for equilibration. The authors state that they use for all samples N_MC<100 sweeps (p.10) – did they observe that this is sufficient for ALL samples to *equilibrate*?

6) Section 3.3 and Appendix A: The authors explain that the advantage of the Sobol sequences they use for disorder realizations is to sample the disorder space more evenly, but is this legitimate: Fig. 2 demonstrates impressively that the usual pseud-random numbers tend to cluster (and leave holes), but isn’t that exactly what leads to a higher probability for strongly coupled clusters (stronger Griffiths singularities), which is suppressed by Sobol sequences?

Requested changes

see report

Recommendation

Publish (easily meets expectations and criteria for this Journal; among top 50%)

---

## Round 1 · Referee Report · Pranay Patil (Referee 2) · 2024-4-24

Strengths

1- Careful analysis of a highly complex problem where there are not many results.

2- Benchmarks of the techniques used have been provided for a simpler 1D version, which is fairly well understood.

3- Improved data analysis methods are implemented, such as more efficient sampling of the disorder phase space, and different extrapolations to the infinite size limit

Weaknesses

1- Due to the large amount of content covered in this manuscript, the reader often finds themselves having to go back to remind themselves of the quantities being considered and the techniques. (See requested changes)

Report

The authors have studied the random transverse field Ising model (RTFIM) using an unbiased quantum Monte Carlo method. The technique is know to be robust and the results are expected to be trustworthy. The analysis of this disorder problem has been carried out by different members of the community over the last few years and conflicting outcomes have been reported. The authors suggest that this may be the result of extreme sensitivity to temperature for this model and have introduced a method for careful data analysis in the low temperature regime. Although the primary target is to understand the 2D RTFIM, the data analysis techniques introduced are benchmarked using the 1D version, which is well understood. The main contribution of their work is to show that temperature effects need to be carefully treated, and that this treatment leads to results consistent with previous works.

Requested changes

1- As there exist conflicting values for the universal exponents from previous studies, it is worth mentioning in detail in the conclusion how this is resolved by quoting the values from previous references and showing how this work makes them consistent.

2- At the end of Sec 4, a small recap would be very useful for the reader. This just needs to mention which quantities are going to be extracted using which method. Same for Sec 5, where this would include a discussion of different exponents, and show the consistency across methods.

3- typo on page 22 : "hardy visible" instead of "hardly visible"

Recommendation

Ask for minor revision

---

## Round 1 · Referee Report · Anonymous (Referee 3) · 2024-4-28

Strengths

A broad review of the current state of the problem

Weaknesses

The manuscript presents extended computational results for the quantum Ising model, however it is not evident that they bring in the essential new physics or methods.

Report

In the manuscript the random quantum Ising model has been examined using the stochastic series expansion quantum Monte Carlo method supplemented by the data analysis for finite-size systems. In the introduction the authors provided a rather broad review of the current state of the problem. It is clear that the study of the random quantum spin models is quite a hard problem with not many rigorous results, and where the results may strongly depend on the type of disorder.
The current study is focused on the accurate study of the critical properties of the model with two types of disorder: bond- and field-disorder, and bond-disorder only. The authors show their expertise in the study of the disordered systems using various methods to achieve the precise calculation of the critical points and exponents. From that point of view, the manuscript seems a bit technical.

Requested changes

The paper contains a lot of material, and I think that its organization could be improved:

1) In Sec.2 the random transverse-field Ising model is introduced, but an accurate description of the averaging over disordered realization is missing here. Therefore, e.g. the definition of the magnetization in Eq.(3) is ambiguous. The right-hand side does not contain thermodynamic or/and random averaging.
Then, in Sec.3.3 the authors introduced Sobol sequences for the disorder average and gave the approximate Eq.(21), while the definition of the disorder averaging was not provided before.

2) In Sec.4.2 the magnetization $m^2(h,L)$ is not defined. Therefore, it is not clear if it is just thermally averaged and also randomly averaged quantity.

Recommendation

Ask for minor revision

---

## Round 1 · Referee Report · Anonymous (Referee 4) · 2024-4-29

Strengths

1- comparison of different finite-size-scaling techniques for disordered transverse-field Ising models
2- good level of detail in describing technical aspects
3- very transparent description of the difficulties of each technique, in particular concerning the temperature convergence
4- valuable for anyone who is interested in numerics for disordered systems or wants to start numerical research in this direction

Weaknesses

1- with its strong focus on technical rigor, the physical message sometimes remains hidden

Report

The authors apply the exact stochastic series expansion quantum Monte Carlo method to the random transverse field Ising model in one and two dimensions. The authors compare different techniques for finite-size scaling and illustrate their advantages/difficulties. The methods are tested for the 1D system, which is much better understood than the 2D case, and then applied to the 2D case to calculate critical couplings/exponents.

I find this numerical study very valuable, as it compares different finite-size-scaling techniques, discusses their strength/weaknesses, and shows what is necessary to arrive at consistent results. One can clearly see that it takes a lot of work to test all of these techniques and analyze the temperature convergence appropriately.

Given the strong technical aspect of this paper (which is very valuable for numerical studies), the physical insights are not always as clear to me. In particular, the authors state that they "resolve several inconsistencies in existing literature". Maybe it would be helpful to highlight these aspects again in the conclusions, as this is one of the parts which is most likely remembered by the reader. Then, it would be easier to find these parts again in the main text.

If I understand correctly, one of the main problems for the 2D case is that results from the sample-replication method are seemingly not consistent with the Binder ratios. Only if one takes the slow temperature and size convergence into account, these discrepancies get resolved. Is this correct?

All in all, I think this paper is of high quality and fulfils the publication criteria of SciPost Physics, once the authors have included my requested changes.

Requested changes

1- On page 4, between Eqs. (2) and (3), the authors write $h\ll \langle J \rangle$ or $h\gg \langle J \rangle$. The meaning of the expectation value has not been defined. Later $\langle \rangle$ has been defined as the thermal average for a given disorder configuration. Do the authors mean the disorder average here, which they denote by $[]$? The same notation also appears again on page 12.

2- On page 6, the definitions of the different parts of the Hamiltonian are slightly confusing. In Eqs. (8) and (9), the indices fulfill $i,j>1$ and $\mathcal{H}_{i,j}$ is only defined for $i\neq j$, because $\mathcal{H}_{i,i}$ includes a different part of the Hamiltonian. I'm also wondering what the constant $c$ in Eq. (7) is needed for, as all the necessary shifts are already included in $\mathcal{H}_{i,j}$. One should also mention at this point that $\mathcal{H}_{0,0}$ is not included in the Hamiltonian, as it is only needed for the fixed-length operator string (as mentioned in Sandvik's original paper).

3- Please check Eq. (19) again. From a quick dimensional analysis, it seems to me that the factor of $1/N^2$ should be $1/N$ and that the prefactor $1/\mathcal{L}$ should be something like $(\beta / \mathcal{L})^2$.

4- In Eq. (20), $\sigma^z_{i,p}$ is not defined.

5- Is Fig. 1 for a single disorder configuration? If yes, one could mention this in the caption.

6- If possible, it would be nice to include figures close to where they are first mentioned in the text. For example, one has to scroll quite a bit through the paper to find Figs. 1 or 3. I understand that this is not always possible, but it would improve readability.

7- I do not find the definition of Eq. (22) very clear, probably because I am not familiar with quasi-random numbers. What is $\lambda_s$?

8- Is there a reason why the periodic boundary conditions are chosen like this? For example, why is 7 connected vertically to 10 and not to 1? Maybe it is worth mentioning the advantage of this?

9- The expansion in Eq. (31) would be much clearer, if the authors referred to the scaling form of Eq. (23) .

10- In the caption of Fig. 10, the abbreviation RTFIC has not been defined, but is also never used anywhere else in the paper.

11- When first looking at Fig. 15, I was wondering why the critical values were not marked in the figure or mentioned in the caption. The caption and the text kind of implied to me that $d/z'=0$ marks the critical value. It took me some time to understand that there are significant finite-size/temperature effects, which are only discussed on the following pages. Maybe it is worth already adding a sentence here, which gives the reader a hint that this issue will be solved below.

12- I know that this is already mentioned in the main text, but I think it would be useful to add to the caption of Fig. 17 that the polynomial fits are of order 1 to 5. Do they appear in sequential order in the figure?

Recommendation

Ask for minor revision

---

## Round 2 · Referee Report · Heiko Rieger (Referee 1) · 2024-6-7

Report

As I stated in my first report the journal criteria for paublication are fulfilled.
The authors complied with all points of my first reprort and I thus recommend publication in its present form.

Recommendation

Publish (surpasses expectations and criteria for this Journal; among top 10%)

---

## Round 2 · Referee Report · Anonymous (Referee 3) · 2024-6-10

Report

In the revised version the authors made the correction according to my remarks. I recommend the paper for publication.

Recommendation

Publish (meets expectations and criteria for this Journal)

---

## Round 2 · Referee Report · Pranay Patil (Referee 2) · 2024-6-10

Report

The authors have considered the modifications requested in a satisfactory manner

Recommendation

Publish (easily meets expectations and criteria for this Journal; among top 50%)

---

## Round 2 · Referee Report · Anonymous (Referee 4) · 2024-6-18

Report

The authors have considered all of my suggestions and answered all of my questions satisfactory. Therefore, I support publication in SciPost Physics.

Recommendation

Publish (easily meets expectations and criteria for this Journal; among top 50%)

---

## Round 2 · Author Response

Warnings issued while processing user-supplied markup:

  • Inconsistency: plain/Markdown and reStructuredText syntaxes are mixed. Markdown will be used.
    Add "#coerce:reST" or "#coerce:plain" as the first line of your text to force reStructuredText or no markup.
    You may also contact the helpdesk if the formatting is incorrect and you are unable to edit your text.

Reply to Report 1

We thank Heiko Rieger for his thorough examination of our manuscript and the positive evaluation. Below we address the individual proposed changes raised by the referee:

  1. To my knowledge, there is a zero-temperature version of SSE (T=0 SSE) which can be applied to the TFIM (Section 1.4.2 in Stochastic series expansion quantum Monte Carlo by Roger G. Melko). As discussed thoroughly by the authors, an extrapolation to T=0 (especially for the RTFIM) requires one to approach very low temperature, so it would be advantageous if such a T=0 SSE can capture T=0 properties without extrapolation. It would be useful if the authors could discuss briefly whether T=0 SSE captures the T=0-properties of the random TFIM or not, or whether the T=0 SSE is hard or impossible to be applied in the present case.

Our answer: T=0 SSE is a different (and less prominent) approach projecting a trial state to the ground state by applying the Hamiltonian several times to it. Since the structure of the algorithm and Monte Carlo updates is similar to the $T \neq 0$ SSE, we would expect the expansion order $m$ to be comparable to our expansion order $n \sim \beta$. Furthermore we could no longer connect the convergence with the physical quantity temperature. However, since we did not implement this method yet, we are not sure, whether the $T=0$ algorithm might improve the convergence. To investigate this further is beyond the scope of this manuscript, but we plan to test the differences between both methods in the future.

  1. The SDRG for the two-dimensional RTFIM (ref. [35] in the paper) predicts that critical exponents of h-fix and h-box are equal, but ref. [40] concludes, also numerically with QMC, that they are different. Although the authors focus on the h-box distribution, exclusively, it would be useful if they found indications pointing in the same direction as in [40] or not.

Our answer: We included in the revised version an evaluation of the 2D h-fix model. We found that the critical exponents are more prone to finite-size effects than the exponents of the h-box model. From the investigation on rather small system sizes we cannot make a clear statement, whether we see a crossover from pure to IDFP exponents or a convergence towards another set of exponents.

  1. p.4: “… rare regions with finite clusters with very special disorder configurations …” I wonder why the authors denote these configurations as “very special”: the common understanding is that these configurations comprise strongly coupled clusters, i.e. clusters (compact or fractal) that are locally in the ordered phase.

Our answer: By “very special” we tried to generalise the phenomena of the locally ordered regions in the disordered phase caused by large J-values in a cluster or the opposite in the ordered phase, where ordered regions can be separated from the rest of the lattice by a boundary of weakly coupled spins. We exchanged the term "very special disorder configurations" by "strongly coupled spin clusters" in the revised version. *

  1. p.5: “The critical exponent of the dynamical scaling is given by psi=1/2” Since usually the dynamic exponent is denoted as z I would more precisely write “… of the activated dynamic scaling …”

Our answer: We included the suggested change in the revised version.

  1. p.9: “causing different disorder configurations to converge in temperature at different $\beta$ values”: I wonder whether similar convergence variations appear in the number of necessary MCS, N_MC, for equilibration. The authors state that they use for all samples N_MC<100 sweeps (p.10) – did they observe that this is sufficient for ALL samples to equilibrate?

Our answer: The necessary $N_MC$ is connected to the autocorrelation time. We would also expect this to vary within disorder configurations. However, as for the temperature, we cannot control the equilibration for each sample individually, but check it for the averaged quantities, which converged very well. In general we experience, that the convergence in temperature is more challenging than the convergence in Monte Carlo sweeps.

  1. Section 3.3 and Appendix A: The authors explain that the advantage of the Sobol sequences they use for disorder realizations is to sample the disorder space more evenly, but is this legitimate: Fig. 2 demonstrates impressively that the usual pseud-random numbers tend to cluster (and leave holes), but isn’t that exactly what leads to a higher probability for strongly coupled clusters (stronger Griffiths singularities), which is suppressed by Sobol sequences?

Our answer: Maybe there is a misunderstand regarding Fig. 2. Each point in Fig. 2 represents one disorder configuration containing only 2 random variables, i.e. the coordinates in the plane (e.g. this could represent the disorder configuration space of a 2-spin system with bond disorder, where the x-axis represents $J_1$ and the y-axis $J_2$). Still, there could be very extreme disorder configurations if the point is close to one of the corners of the plane. In our understanding using Sobol sequences only improves that the portion of rare configurations in a finite set of disorder configurations is closer to the right portion than for random numbers. We tried to make it clearer by adding an additional remark in the caption of Fig. 2.

Reply to Report 2

We thank Pranay Patil for his thorough examination of our manuscript and the positive evaluation. Below we address the individual proposed changes raised by the referee:

  1. As there exist conflicting values for the universal exponents from previous studies, it is worth mentioning in detail in the conclusion how this is resolved by quoting the values from previous references and showing how this work makes them consistent.

Our answer: We added concluding statements that summarize our results on the critical exponents and if they are consistent with previous results or not.

  1. At the end of Sec 4, a small recap would be very useful for the reader. This just needs to mention which quantities are going to be extracted using which method. Same for Sec 5, where this would include a discussion of different exponents, and show the consistency across methods.

Our answer: In the revised version, we included a brief recap of all data analysis methods in the beginning of Sec. 5, where we summarise which method is used in the following to extract which quantity.

  1. typo on page 22 : "hardy visible" instead of "hardly visible"

Our answer: We corrected the typo in the revised version.

Reply to Report 3

We thank the Referee for their thorough examination of our manuscript and the positive evaluation. Below we address the individual proposed changes raised by the referee:

  1. In Sec.2 the random transverse-field Ising model is introduced, but an accurate description of the averaging over disordered realization is missing here. Therefore, e.g. the definition of the magnetization in Eq.(3) is ambiguous. The right-hand side does not contain thermodynamic or/and random averaging. Then, in Sec.3.3 the authors introduced Sobol sequences for the disorder average and gave the approximate Eq.(21), while the definition of the disorder averaging was not provided before.

Our answer: In the revised version, we introduce the thermal and disorder average in Sec. 2. Furthermore we clarified that the magnetisation defined in Eq. (3) is not the order parameter we actually look at, but the averaged squared magnetisation.

  1. In Sec.4.2 the magnetization $m^2(h,L)$ is not defined. Therefore, it is not clear if it is just thermally averaged and also randomly averaged quantity.

Our answer: We included the missing $\langle ... \rangle$ in Sec. 4.2 and elsewhere in the paper, where it was missing.

Reply to Report 4

We thank the Referee for their thorough examination of our manuscript and the positive evaluation. Below we address the individual proposed changes raised by the referee:

  1. On page 4, between Eqs. (2) and (3), the authors write $h \ll \langle J \rangle$ or $h \gg \langle J \rangle$. The meaning of the expectation value has not been defined. Later $\langle ... \rangle$ has been defined as the thermal average for a given disorder configuration. Do the authors mean the disorder average here, which they denote by $[...]$? The same notation also appears again on page 12.

Our answer: Actually in this case both definitions $\langle ... \rangle$ and $[...]$ are not suitable in the way we defined them, since in this context simply the mean of the $J_{i,j}$ was intended. We rephrased the description of the two phases in the revised version omitting the misleading notation.

  1. On page 6, the definitions of the different parts of the Hamiltonian are slightly confusing. In Eqs. (8) and (9), the indices fulfill $i,j > 1$ and $\mathcal{H}_{i,j}$ is only defined for $i \neq j$, because $\mathcal{H}_{i,i}$ includes a different part of the Hamiltonian. I'm also wondering what the constant $c$ in Eq. (7) is needed for, as all the necessary shifts are already included in $\mathcal{H}_{i,j}$. One should also mention at this point that $\mathcal{H}_{0,0}$ is not included in the Hamiltonian, as it is only needed for the fixed-length operator string (as mentioned in Sandvik's original paper).

Our answer: We included the suggested changes regarding the definition of the operators $\mathcal{H}_{i,j}$. The factor $c$ just compensates for the shifts made in $H_{i,j}$ compared to the original Hamiltonian defined earlier.

  1. Please check Eq. (19) again. From a quick dimensional analysis, it seems to me that the factor of $1/N^2$ should be $1/N$ and that the prefactor $1/\mathcal{L}$ should be something like $(\beta/\mathcal{L})^2$.

Our answer: We corrected the wrong pre-factors in the revised version.

  1. In Eq. (20), $\sigma_{i, p}^z$ is not defined.

Our answer: We added the definition of $\sigma_{i, p}^z$ of in the revised version.

  1. Is Fig. 1 for a single disorder configuration? If yes, one could mention this in the caption.

Our answer: Fig. 1 shows the disorder averaged magnetisation at the critical point. In the revised version we added this information in the caption of the figure.

  1. If possible, it would be nice to include figures close to where they are first mentioned in the text. For example, one has to scroll quite a bit through the paper to find Figs. 1 or 3. I understand that this is not always possible, but it would improve readability.

Our answer: We rearranged the figures manually in the revised version.

  1. I do not find the definition of Eq. (22) very clear, probably because I am not familiar with quasi-random numbers. What is $\lambda_s$?

Our answer: We added the definition of $\lambda_s$ and a brief explanation to the definition of the discrepancy of a set.

  1. Is there a reason why the periodic boundary conditions are chosen like this? For example, why is 7 connected vertically to 10 and not to 1? Maybe it is worth mentioning the advantage of this?

Our answer: The choice how we coupled the doubled system results in an isotropic lattice with alternating pattern of the original and copied system. E.g. if you cross the boundary horizontally from 15 to 4, you end up crossing from the copied (green) to the original (blue) system. The same should hold when you cross vertically from e.g. 7 to 10. If you would connect 10 to 1, you would cross from the original system to the original system again.

  1. The expansion in Eq. (31) would be much clearer, if the authors referred to the scaling form of Eq. (23).

Our answer: We added the suggested reference in the revised version.

  1. In the caption of Fig. 10, the abbreviation RTFIC has not been defined, but is also never used anywhere else in the paper.

Our answer: We corrected the abbreviation in the caption of Fig. 10 in the revised version.

  1. When first looking at Fig. 15, I was wondering why the critical values were not marked in the figure or mentioned in the caption. The caption and the text kind of implied to me that $d/z'= 0$ marks the critical value. It took me some time to understand that there are significant finite-size/temperature effects, which are only discussed on the following pages. Maybe it is worth already adding a sentence here, which gives the reader a hint that this issue will be solved below.

Our answer: In the revised version, we added the information in the text and Fig. 16 (previously Fig. 15), that the position of $d/z'= 0$ is affected finite-size and temperature effects and indicate that this is further investigated in the next section.

  1. I know that this is already mentioned in the main text, but I think it would be useful to add to the caption of Fig. 17 that the polynomial fits are of order 1 to 5. Do they appear in sequential order in the figure?

Our answer: Yes they did, but served anyway just as a guide to the eye. For the revised version, we calculated the same quantity at lower temperatures and updated the figure. Instead of a logarithmic scale, we decided to show the finite-size and temperature dependent critical points over the temperature T and omitted the polynomial fits.

---

## Round 2 · List of Changes

Line numbers are indicated for orientation.

Introduction
- l.60 Changed 'Model' to 'model'
- l.91-92 Added 'in the vicinity of the critical point' and additional citation

Random transverse-field Ising model
- Eq. 1,2 Changed the sign of J_{i,j} to be consistent with Sec. 3
- l.123 Changed the sign of J_{i,j} to be consistent with Sec. 3
- l.124-126 Removed misleading notation of average over bond-strengths, reformulated sentence
- l.130-136 Introduced thermal and disorder average here to clarify, which order parameter we actually consider
- Eq. 4 Introduced absolute values since sign of J_{i,j} changed in Eq. 1
- l.150 Added 'in the vicinity of the critical point' and additional citation
- l.155 Changed 'finite clusters with very special disorder configurations' to 'strongly coupled spin clusters'
- l.172 Inserted "activated"
- l.176 Changed $\nu_{s/w}$ to $\nu_{\mathrm{s}/\mathrm{w}}$

Stochastic series expansion quantum Monte Carlo
- Eq.7,10 Changed upper bound of the second sum to be consistent with Eq. 8,9
- l.214-215 Added sentence to clarify, what values the indices i,j can be and that the trivial operator is not part of the Hamiltonian
- Eq.11,12 Inserted curly brackets around S_n and S_\mathcal{L} to clarify that we sum over sets of sequences
- Eq. 19 Corrected prefactors
- l.267-268 Added explanation to Eq. 20
- l.292-293 Added "and equilibration" and "separately"
- Fig.1 Added the information in the caption, that the averaged magnetisation at the critical point is shown; Added correct notation for the averages in the labels of the figure.
- l.314 Removed the definition of disorder average here, since it is now introduced earlier in Sec. 2
- l.315-316 Added "in the vicinity of the critical point, which is the region we are interested in." and additional citation
- Fig. 2 Clarified in the caption that the figure would correspond to a two-spin system with h-fix disorder
- l.345-348 Added additional explanations to the definition of the discrepancy in Eq. 22

Data analysis for disordered systems
- Fig. 4 Clarified in the caption, why the periodic boundary conditions are chosen in the way depicted
- l.409-410 Added brackets denoting thermal average to the magnetisation
- Eq. 27 Added brackets denoting thermal average to the magnetisation
- l.413 Removed misleading notation of average over bond-strengths
- Fig. 5 Added information in the caption, which model is shown.
- l.443, 445 Referred back to Eq. 23 to make Eq. 31 clearer; identified \omega in Eq. 23 with \beta in Eq. 31
- l.452 Added additional citation

Results
- l.471-483 Summarized data-analysis methods and which quantity is determined with which method
- l.484-485 Added reference to raw data
- Eq. 37 Introduced absolute values since sign of J_{i,j} changed in Eq. 1
- l.547 Changed $\nu$ to $\nu_{\mathrm{w}}$
- Fig. 9 Added brackets denoting thermal and disorder average in y-labels
- l.553 Added brackets denoting thermal and disorder average to the magnetisation
- l.557 Changed 'hardy' to 'hardly'
- Fig. 10 Added brackets denoting thermal and disorder average to the magnetisation in the caption; changed 'RTFIC' to 'RTFI chain'
- l.579-591 Reformulated description of systems investigated since we also include data of the 2D h-fix model (regarding the Binder ratios and averaged magnetisation) in the revised version
- Fig. 11 Added plot showing the intersections of Binder ratios for the h-fix model. Mentioned the h-fix model in the caption
- l.594-598 Added results from investigation of h-fix model
- Fig. 13 Changed 'by Ref. [40]' to 'using the intersections of Binder ratios' in the caption; used the value, we determined using the Binder analysis instead of literature values for the plot.
- l.617-622 Changed formulation since we now actively investigate h-fix with all primary methods instead of just for comparison; changed 'hardy' to 'hardly'
- l.624 Added brackets denoting thermal and disorder average to the magnetisation
- l.625-640 Added results for the 2D h-fix model and discussed them
- l.643 Removed 'the'
- Fig. 14 Added plot showing the same data for the h-fix model; Added brackets denoting thermal and disorder average in y-labels; Changed caption accordingly
- Fig. 15 New figures that shows the L-dependency of the critical exponents
- Fig. 16 Added in caption 'whose position is affected by strong finite-size and temperature effects (see Sec. 5.3)'
- l.652-655 Mentioned finite-size and temperature effects
- l.655-658 Restructured observation for 2D h-fix model
- l.662-663 Removed 'Here, we did not investigate the critical exponents for the 2D Hfix model'
- l.696-703 Reformulated interpretation of the data for d/z', since we produced new data with significantly lower temperature and plotted them over 1/T instead of on a logarithmic scale
- Fig. 17 Using new data with significantly lower temperature and changed axis in the right plot; Left out extrapolations; Changed information accordingly in the caption

Conclusion
- l.729-734 Included discussion on findings regarding critical exponents in 2D with different type of disorder

Acknowledgements
- l.748 We thank Federico Becca for the fruitful e-mail exchange.
- l.759-760 Added reference to raw data

Appendix A: Verification
- Fig. 20 Corrected the dimension of the disorder configuration space (s=3*L^2 for the square lattice)

Appendix B: Jordan-Wigner calculation of correlation functions for the 1D-RTFIM
- l.835-836 Removed 'transverse-field Ising model', since abbreviation is already introduced earlier
- Eq. 39,40 Aligned equations
- Eq. 42 Corrected typo, moved part of the equation to next line
- l.863 Inserted \dots in brackets

Appendix C: Further finite-size scaling methods for disordered systems
- l.931 Changed 'verfication' to 'verification'
- l.934-935 Changed 'SSE' to 'the SSE QMC method'
- l.962 Inserted 'Following Eq. (23)'
- l.963,965 Added brackets denoting thermal and disorder average to the magnetisation
- Fig. 27 Added brackets denoting thermal and disorder average to the magnetisation in the y-label of the left figure
- l.990 Changed 'and the non-universal constants $c$ and $d$' to 'and $c$ and $d$ are non-universal constants'

---

## Editorial Decision

published